

# Assessing Effects of Climate and Technology Uncertainties in Large Natural Resource Allocation Problems

Jevgenijs Steinbuks[1], Yongyang Cai[2], Jonas Jaegermeyr[3,4,5], and Thomas W. Hertel[6]

[1]Office of the Chief Economist, Infrastructure Vice Presidency, The World Bank, Washington, DC, USA
[2]Department of Agricultural, Environmental and Development Economics, The Ohio State University, Columbus, OH, USA
[3]Department of Computer Science, University of Chicago, Chicago, IL, USA
[4]NASA Goddard Institute for Space Studies, New York, NY, USA
[5]Climate Impacts and Vulnerabilities, Potsdam Institute for Climate Impact Research, Potsdam, Germany
[6]Center for Global Trade Analysis, Purdue University, West Lafayette, IN, USA

**Correspondence:** Jevgenijs Steinbuks (jsteinbuks@worldbank.org)

**Abstract.** The productivity of the world's natural resources is critically dependent on a variety of highly uncertain factors, which obscure individual investors and governments that seek to make long-term, sometimes irreversible investments in their exploration and utilization. These dynamic considerations are poorly represented in disaggregated resource models, as incorporating uncertainty into large-dimensional problems presents a challenging computational task. In this paper, we apply the SCEQ algorithm (Cai and Judd, 2021) to solve a large-scale dynamic stochastic global land resource use problem with stochastic crop yields due to adverse climate impacts and limits on further technological progress. For the same model parameters, the range of land conversion is considerably smaller for the dynamic stochastic model as compared to deterministic scenario analysis. The scenario analysis can thus significantly overstate the magnitude of expected land conversion under uncertain crop yields.

## 1 Introduction

Understanding the future allocation of the world's natural resources is an important research problem for environmental scientists and economists. This involves a thorough understanding of a complex interplay of different factors, including, among others, continuing population increases, shifting diets among the poorest populations in the world, increasing production of renewable energy, including biofuels, and growing demand for ecosystem services, including forest carbon sequestration (Foley et al., 2011). The problem is further confounded by faster than expected climate change, which is altering the biophysical environment of agriculture and forestry. Moreover, highly uncertain future productivities and valuations of ecosystem services, coupled with medium- to long-term irreversibilities in the extraction of nonrenewable or partially renewable resources, such as long-growth natural forests, give rise to a challenging problem of decision-making under uncertainty.

While there is a large body of research analyzing the problem of natural resource extraction and utilization under uncertainty theoretically or using stylized computational models (see, e.g., Miranda and Fackler (2004), Tsur and Zemel (2014) and references therein), quantifying the effects of uncertainty on natural resource use in a more realistic setting remains a challenging





problem. This is because natural resource allocation problems, like environmental policy problems in general, involve highly nonlinear structure and damage functions, important irreversibilities, and long time horizons (Pindyck, 2007). Computational integrated models of economy and environment are the standard workhorse mechanisms for modeling the long-term alloca-

tion of the world's natural resources, including particularly difficult land use problems (see, e.g., Füssel (2009), Schmitz et al. (2014), Nikas et al. (2019), and references therein). These models have the important advantage of detailed spatial and sectoral (particularly, energy and agricultural sector) coverage, which allows them to capture a broad range of responses to changes in demand and supply factors affecting the utilization of natural resources. However, given high computational complexity of these models, they are typically either static or based on myopic expectations, whereby decisions about production, consump-

tion, and resource extraction and conversion are made only on the basis of information in the period of the decision (Babiker et al., 2009). These models, therefore, have limited ability to address important intertemporal questions such as, for example, a dynamic trade-off between conservation, carbon sequestration, and renewable offsets for fossil fuels. Among the few forward-looking, dynamic economy and environment models, none explicitly incorporates uncertainty into the determination of the optimal path of natural resource use.[1] This is because introducing uncertainty into these models is confined by an array of

computational obstacles that are very difficult (e.g., high dimensionality and kinks caused by occasionally binding constraints), if not impossible, to address using standard numerical methods such as projection methods and value function iteration (see, e.g., Judd 1998; Miranda and Fackler 2004; Cai and Judd 2014; Cai 2019). To the extent that uncertainty in these models is considered, this is only through parametric or probabilistic sensitivity analysis or the use of alternative scenarios. Therefore the high-dimensional resource use models have not effectively dealt with optimal extraction and conversion decisions along the

uncertain path of key drivers affecting resource allocation in the face of costly reversal of conversion decisions.

In this study, we seek to address this important limitation of the economy-environment modeling of natural resource use. In doing so, we build on recent advances in computational economics and operations research. Cai et al. (2017) introduced a nonlinear certainty equivalent approximation method (NLCEQ) for solving large-scale infinite horizon stationary dynamic stochastic problems and demonstrated how this method could be used to achieve an accurate solution to a stylized stationary

dynamic stochastic land use problem. The original NLCEQ method, however, is ill-suited for solving most environmental and resource economics problems. This is because stochastic problems of utilization of natural resources feature nonstationary stochastic trends, such as climate or technological trajectories, and some never converge to a stationary state. Building on the original NLCEQ work Cai and Judd (2021) introduced a simulated certainty equivalent approximation method (SCEQ), which efficiently solves nonstationary dynamic stochastic problems, including those with high dimensionality and occasionally

binding constraints. Cai and Judd (2021) showed that the SCEQ method is highly accurate and achieves stable numerical solutions for dynamic stochastic problems in economics.

We apply the SCEQ method to solve a large-scale dynamic stochastic model, focusing on the optimal global land use allocation problem. This is a highly complicated resource use problem that features multiple cross-sectoral and dynamic trade-offs. Specifically, we apply the method to a global land use model nicknamed FABLE (Forest, Agriculture, and Biofuels

---

[1]Several recent studies, e.g., Cai and Lontzek (2019) have successfully integrated uncertainty about economic and climate outcomes in a stochastic integrated assessment climate-economy framework. For a review of this related literature, see Cai (2019, 2021).





in a Land use model with Environmental services) in the face of uncertainty. FABLE is a dynamic, forward-looking global multi-sectoral partial equilibrium model designed to analyze the evolution of global land use over the coming century. Prior applications of that model (Steinbuks and Hertel, 2013; Hertel et al., 2013, 2016; Steinbuks and Hertel, 2016) analyze the competition for scarce global land resources in light of the growing demand for food, energy, forestry, and environmental services, and evaluate key drivers and policies affecting global land use allocation. All these applications, however, assume perfect foresight and treat uncertainty in a parametric fashion, thus ignoring the impact of future uncertainties on the optimal allocation of global land use.

By way of illustration, we choose to focus on uncertainty emanating from crop productivity over the next century. Along with energy prices, regulatory policies, and technological change in food, timber, and biofuels industries, this is one of four core uncertainties affecting competition for global land use (Steinbuks and Hertel, 2013). To quantify the uncertainty in agricultural yields, we construct a stochastic crop productivity index that captures two key uncertainty sources: technological progress and global climate change (Lobell et al., 2009; Licker et al., 2010; Foley et al., 2011).[2] Following Rosenzweig et al. (2014), we use projections from climate and crop simulation models, as well as the survey of recent agro-economic and biophysical studies to calibrate the index.

We simulate the results of the perfect foresight model under different scenarios of the crop productivity index, focusing our attention on the current century. We then compare and contrast them with the results of the dynamic stochastic model, where the uncertainty in crop yields is brought to the model's optimization stage. When the uncertainty in crop productivity is incorporated into the model, we see an additional redistribution of land resources aimed at offsetting the impact of potentially lower yields. Owing to intertemporal substitution, some of that redistribution takes place even in the absence of the actual changes in the states of climate or technology affecting crop yields. Moreover, the range of these alternative optimal paths of cropland is considerably smaller than the magnitude of possible land conversion resulting from the scenario analysis based on the deterministic model. This result indicates that the scenario analysis may significantly overstate the magnitude of expected agricultural land conversion under uncertain crop yields.

Our study contributes to the growing literature that analyzes the intertemporal allocation of land and other natural resources under uncertainty and irreversibility constraints. Most of the literature focuses on a particular type of resource or sector, where intertemporal issues are significant and cannot be ignored. One example of this literature is forestry management in the context of uncertain fire risks and climate mitigation policies (Sohngen and Mendelsohn, 2003, 2007; Daigneault et al., 2010). Another example is natural land conservation decisions under irreversible biodiversity losses (Conrad, 1997, 2000; Bulte et al., 2002; Leroux et al., 2009). While these models are undoubtedly helpful for understanding the broad implications of uncertainty on the intertemporal allocation of land resources, they fail to account for the effect of uncertainty in the supply and demand drivers on the optimal allocation of land resources in the long run. Our study is perhaps most closely related to the recent work of Lanz et al. (2017) who develop a two-sector stochastic Schumpeterian growth model with the endogenous allocation of global

---

[2]Climate change will likely affect the productivity of other land resources, such as forestland. Several recent modeling studies (see, e.g., Tian et al. (2016) and references therein) have suggested that climate change is likely to result in higher forest growth and greater timber yields, as well as in more forest dieback, with the net effects varying over time and space. Incorporating these effects is beyond the scope of this study and is left for future research.





land use. They find, like our paper does, that optimal allocation of global land use requires more cropland conversion when the uncertainty in agricultural productivity is present. Lanz et al. (2017), however, focused on endogenous population dynamics, labor allocation, and technological progress, whereas our paper is concerned about the endogenous allocation of multiple types
of land use and corresponding land-based goods and services. Our paper also advances methodological grounds by applying a more advanced algorithm that overcomes computational difficulties in solving multidimensional stochastic land use models, which made Lanz et al. (2017) significantly simplify their model by assuming that their binary shocks occur only in three time periods.

## 2   Stochastic FABLE model

This section presents a modeling framework for analyzing nonlinear dynamic stochastic models of natural resource use with multiple sectors, in which preferences, production technology, resource endowments, and other exogenous state variables evolve stochastically over time according to a Markov process with time-varying transition probabilities. The constructed model belongs to the class of stochastic growth models with multiple sectors studied in Brock and Majumdar (1978), Majumdar and Radner (1983), and Stokey et al. (1989) among others.

Specifically, we develop a stochastic version of a global land use model nicknamed FABLE (Forest, Agriculture, and Biofuels in a Land use model with Environmental services), a dynamic multi-sectoral model for the world's land resources over the next century (Steinbuks and Hertel, 2012, 2016). This model brings together recent strands of agronomic, economic, and biophysical literature into a single, intertemporally consistent, analytical framework, at the global scale. FABLE is a discrete dynamic partial equilibrium model, where the population, labor, physical and human capital, and other variable inputs are assumed to
be exogenous. Total factor productivity and technological progress in non-land intensive sectors are also predetermined. The model focuses on the optimal allocation of scarce land across competing uses across time and solves the dynamic paths of alternative land uses, which together maximize global economic welfare.

    The FABLE model accommodates a complex dynamic interplay between different types of global land use, whereby the societal objective function places value on processed crops and livestock, energy services, timber products, ecosystem services,
and other non-land goods and services (Figure 1). There are three accessible primary resources in this partial equilibrium model of the global economy: land, liquid fossil fuels, and other primary inputs, e.g., labor and capital (see the bottom part of Figure 1). The supply of land is fixed and faces competing uses that are determined endogenously by the model. They include unmanaged forest lands - which are in an undisturbed state (e.g., parts of the Amazon tropical rainforest ecosystem), agricultural (or crop) land, pasture land, and commercially managed forest land. As trees of different age have different timber
yields and different propensities to sequester carbon, the model keeps track of various tree vintages in managed forests, which introduces additional numerical complexity for solving the model. We don't keep track of vintages for natural lands and assume they are primarily old-grown forests. We ignore other land use types, such as savannah, grasslands, and shrublands, which are largely unmanaged and often of limited productivity. This makes them difficult to incorporate into an economic model of land



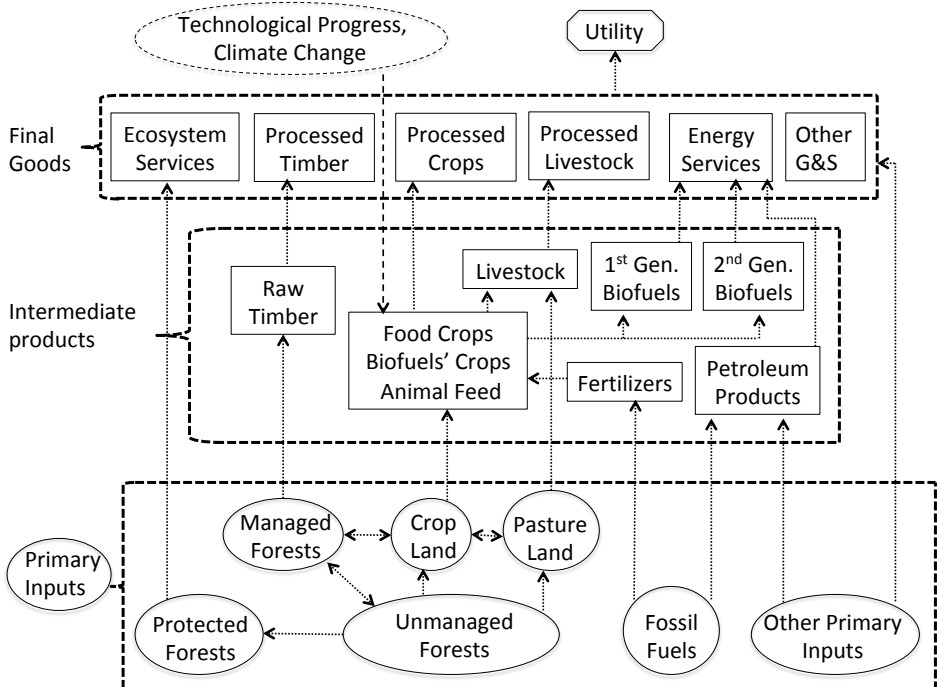

**Figure 1.** Structure of the FABLE Model

Note. State variables are shown as oval shapes. Decision variables are shown as rectangular shapes. The utility function is shown as an octagonal shape. Transition laws are shown as dotted arrows. Stochastic model terms incorporating random processes are shown as dashed shapes or arrows.

use. Consequently, they are typically left out of most contemporary analyses of global land use change (Hertel et al., 2009). We also ignore residential, retail, and industrial uses of land in this partial equilibrium model of agriculture and forestry.

The flow of liquid fossil fuels evolves endogenously along an optimal extraction path, allowing for exogenously specified new discoveries of fossil fuel reserves. Other primary inputs include variable inputs, such as labor, capital (both physical and human), and intermediate materials. The endowment of other primary inputs is exogenous and evolves along a pre-specified global economic growth path.

There are six intermediate inputs used in the production of land-based goods and services in FABLE: petroleum products, fertilizers, crops, liquid biofuels,[3] live animals, and raw timber (see the middle part of Figure 1). Fossil fuels are refined and converted to either petroleum products, that are further combusted, or to fertilizers, that are used to boost yields in the agricultural sector. Cropland and fertilizers are combined to grow crops, that can be further converted into processed food and

---

[3]In FABLE, bioenergy does not include the potential use of biomass in power generation. This limitation is acknowledged in Steinbuks and Hertel (2016, p. 566): "A more serious limitation to this study is our omission of the potential demand for biomass in power generation. Under some scenarios, authors have shown this to be an important source of feedstock demand by mid-century (Rose et al. 2012). However, absent a full representation of the electric power sector, our framework is ill-suited to addressing this issue."





biofuels, or used as animal feed. Specifically, we assume that agricultural land $L_t^{A,c}$ and fertilizers $x_t^{n,c}$ are imperfect substitutes
in production of food crops, $x_t^c$, with specific production technology given by the following constant elasticity of substitution
(CES) function:

$$x_t^c = \theta_t^c \left( \alpha^n \left( L_t^{A,c} \right)^{\rho_n} + (1 - \alpha^n) (x_t^{n,c})^{\rho_n} \right)^{\frac{1}{\rho_n}}, \tag{1}$$

where $\theta_t^c$ is stochastic crop technology index, and $\alpha^n$ and $\rho_n$ are, respectively, the input share and substitution parameters.
Equation (1) captures three key responses within the model to changes in crop technology index: (i) demand response (change in
consumption of food crops), (ii) adaptation on the extensive margin (substitution of agricultural land for other land resources),
and (iii) adaptation on the intensive margin (substitution of agricultural land for fertilizers).

    Biofuels substitute imperfectly for liquid fossil fuels in final energy demand. The food crops used as animal feed and pasture
land are combined to produce raw livestock. Harvesting managed forests yield raw timber, that is further used in timber
processing.

The land-based consumption goods and services take the form of processed crops, livestock, and timber, and are, respectively,
outcomes of food crops, raw livestock, and timber processing. The production of energy services combines non-land energy
inputs (i.e., liquid fossil fuels) with the biofuels, and the resulting mix is further combusted. Finally, all land types have
the potential to contribute to other ecosystem services, a public good to society, which includes recreation, biodiversity, and
other environmental goods and services. To close the demand system, we also include other non-land goods and services
(e.g., manufacturing goods and retail, construction, financial, and information services), which involve 'consumption' of other
primary inputs not spent on the production of land-based goods and services. As the model focuses on the representative agent's
behavior, the final consumption products are all expressed in per-capita terms.

    A complete description of model equations, variables, and parameter values is presented in the appendix.

## 3   Modeling Crop Yield Uncertainty

This section characterizes uncertainty in future agricultural yields over the coming century, which is one of the core uncertainties shown to affect land use in the long run (Steinbuks and Hertel, 2013). Crop yields are subject to two types of uncertainties:
those related to the development and dissemination of new technologies, and those related to changes in the climatic conditions
under which the crops are grown. The former type of uncertainty has until recently dominated the pattern of the evolution of
global crop yields, whereas the latter is becoming an increasingly important factor (Lobell and Field, 2007; IPCC, 2014a).
While it is plausible to hypothesize that accelerating climate impacts may, in turn, induce further technological advances in
an effort to facilitate adaptation to climate change, this hypothesis is not supported by limited empirical evidence (Burke
and Emerick, 2016). Therefore, in this paper, these two sources of uncertainty are treated separately, although they are both
characterized by the use of combined climate and crop simulation models run over a global grid.

    We characterize future uncertainty in yields by constructing a stochastic crop productivity index, $\theta_t^c$, which captures the
evolution of future crop yields under different realizations of uncertainty in crop productivity based on the most recent pro-





jections in the agronomic and environmental science studies. An important characteristic of staple grains yields is that they tend to grow linearly, adding a constant amount of gain (e.g., ton/ha) each year (Grassini et al., 2013). This suggests that the proportional growth rate should fall gradually over time. However, crop physiology dictates certain biophysical limits to the rate at which sunlight and soil nutrients can be converted to the grain. And there is some recent agronomic evidence (Cassman

et al., 2010; Grassini et al., 2013) showing that yields appear to be reaching a plateau in some of the world's most important cereal-producing countries. Cassman (1999) suggests that average national yields can be expected to plateau when they reach 70–80% of the genetic yield potential ceiling. Based on these observations from the agronomic literature, we specify the following logistic function determining the evolution of the crop productivity index over time:

$$\theta_t^c = \frac{\theta_T^c \theta_0^c e^{\kappa_c t}}{\theta_T^c + \theta_0^c \left(e^{\kappa_c t} - 1\right)}, \tag{2}$$

where $\theta_0^c$ is the value of the crop productivity index in period 0, which we calibrate to match observed weighted yields in key staple crops (corn, rice, soybeans, and wheat), $\theta_T^c$ is the crop yield potential in the end of the current century, that is, "the yield an adapted crop cultivar can achieve when crop management alleviates all abiotic and biotic stresses through optimal crop and soil management" (Evans and Fischer, 1999), and $\kappa_c$ is the logistic convergence rate to achieving potential crop yields.

Though the initial value of the crop productivity index is known with certainty, potential crop yields are highly uncertain. We

assume that potential crop yields are affected by a two-dimensional stochastic process of climate and technological shocks, $J_{1,t}$, and $J_{2,t}$, respectively. For the technological shock, $J_{2,t}$, we assume that there are three states of technology: "bad" (indexed by $J_{2,t} = 1$), "medium" (indexed by $J_{2,t} = 2$), and "good" (indexed by $J_{2,t} = 3$). In the optimistic (i.e., "good") state of advances in crop technology, we assume that yields continue to grow linearly throughout the coming century, eliminating the yield gap by 2100. In the "medium" state of technology, rather than closing the yield gap by 2100, average yields in 2100 are just three-

quarters of the yield potential at that point in time. In the "bad" state of technology, there is no technological progress, and the crop yields stay the same as at the beginning of the coming century.

For the climate shock, $J_{1,t}$, we assume it is a Markov chain with five possible states at each time $t$. To construct these states, we use the results of Rosenzweig et al. (2014), who conducted a globally consistent, protocol-based, multi-model climate change assessment for major crops with the explicit characterization of uncertainty. Given the partial equilibrium nature of

FABLE, we cannot directly capture all sources of GHG emissions and, therefore, endogenize their effect on global land use. However, as global land use emissions account for less than a quarter of global GHG emissions (IPCC, 2014b), climate-induced changes in land use will be relatively small to have a major effect on global temperatures.

Based on this assessment, we construct five states that correspond to quintiles of the distribution of different outcomes of four global crop simulation models and five global climate models, with and without $CO_2$ fertilization effects for potential

crop yields by 2100. Under two optimistic states of the world, we observe a 2 and 15 percent increases in potential crop yields relative to the model baseline, respectively, whereby significant $CO_2$ fertilization effects offset the negative effects of climate change. For the next two states, we see a 15 and 19 percent declines in potential crop yields relative to the model baseline whereby $CO_2$ fertilization effects are assumed to be either small or non-existent, and the negative effects of climate change



tend to prevail. Finally, under the most pessimistic state of the world, drastic adverse effects of climate change combined with

the absence of any $CO_2$ fertilization effects result in a 36 percent decline in potential crop yields relative to the model baseline.

Further details of constructing climate and technological states can be found in the appendix.

The path of technological change in crop yields evolves by reversible transitions across these states. The stochastic path of the crop productivity index is then given by

$$A_t = \frac{A_T(J_{1,t}, J_{2,t})A_0 e^{\kappa_c t}}{A_T(J_{1,t}, J_{2,t}) + A_0 \left(e^{\kappa_c t} - 1\right)}, \tag{3}$$

where $A_T(i,j)$ represents the crop productivity index at the terminal time $T$ at the state $J_{1,t} = i$ and $J_{2,t} = j$, for $i = 1, 2, ..., 5$ and $j = 1, 2, 3$. Thus, $A_t$ is a Markov chain, which takes one of 15 possible time-varying values at each time period. This can be seen as a discretization of a mean-reverting process with continuous values and a time trend, but a finer Markov chain with more values can only marginally change our solution. As $A_t$ is completely dependent on $J_{1,t}$ and $J_{2,t}$, it is not a state variable, whereas $J_{1,t}$ and $J_{2,t}$ are both state variables. Having characterized the realizations of crop productivity under alternative

states of the agricultural technology and climate change impacts, we still need to calibrate the transition probabilities for the climate and technology shocks to construct the stochastic crop productivity index. As regards climate shock, the environmental and climate science literature acknowledges some degree of persistence but does not provide much guidance on the transition dynamics between alternative climate states affecting crop yields. In the absence of reliable estimates, for constructing the transition probability matrix of $J_{1,t}$ we assume simple transition dynamics, where each state has a 50 percent probability

of retaining itself next period and a 25 percent probabilities of moving upwards and downwards to an adjacent state. As regards the technology shock, since we do not have historical data for the evolution of agricultural technology, we assume that technological advances in agriculture follow a similar trend to advances in the rest of the economy, and use the probability transition matrix of $J_{2,t}$ estimated by Tsionas and Kumbhakar (2004) for a comprehensive panel of 59 countries over the period of 1965–1990. These estimates correspond to a 20 percent probability of the "bad" technological state, 56 percent of the

"medium" state, and 24 percent of the "good" state. The transition probability matrices of $J_{1,t}$ and $J_{2,t}$ are shown in appendix.

Figure 2 shows the deterministic-baseline path (the solid line) used in the perfect foresight model and the range of the stochastic crop productivity index based on 1,000 simulation paths over the the entire 21st century, with additional summary statistics presented in appendix. The simulations start at the "medium" states of climate and technology in the initial year. The deterministic-baseline path is calculated by taking expectations of the stochastic crop productivity index conditional on the

initial "medium" states (equation 3). It also takes the same values as the median path (the "o" line) of simulations, whereby the climate and technological states are kept at "medium", while the average line (the "+" line) deviates a bit after the year 2070. At every time $t$, there are 1,000 realized values of $A_t$ among which there are only 15 different values. The 10% and 90% quantile lines (the dashed and dash-dotted lines) represent the 10% and 90% quantiles of these 1,000 simulated values of $A_t$ at time $t$, so they are not realized sample paths, but Figure 2 also displays one realized sample path of $A_t$ which is the dotted line.





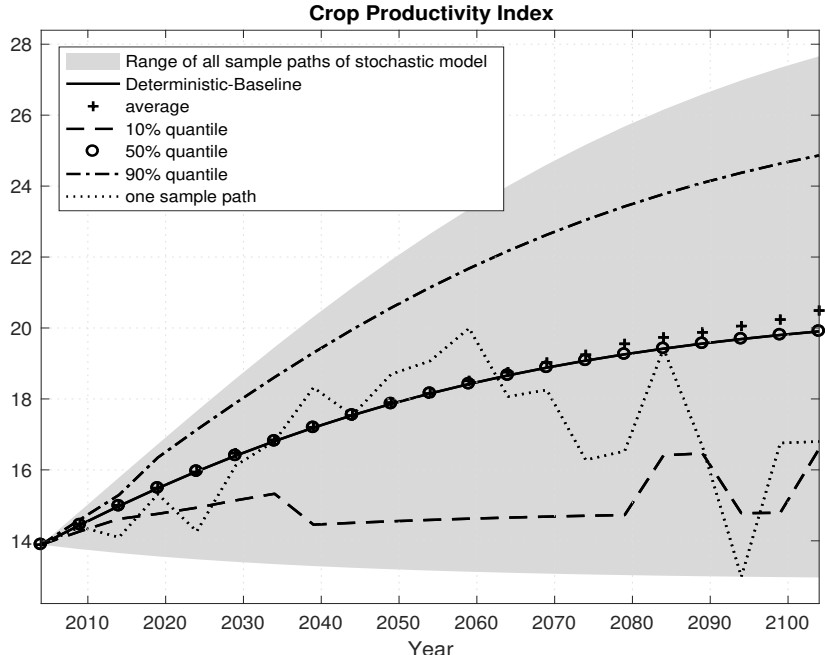

**Figure 2.** Crop Productivity Index

## 225  4  Model Results

We solve the model using the SCEQ method (Cai and Judd, 2021), which is also briefly described in the appendix. Below we describe the results of the impact of crop yield uncertainty on the optimal path of global land use based on the dynamic stochastic model simulations. While it is unlikely that the world's land will be optimally allocated in the coming century, knowledge of this path can provide important insights into how uncertainty and irreversibility shape the desired path for global land use decisions. We solve the model over 400 years with 5-year time steps and present the results for the first 100 years to minimize the effect of terminal period conditions on our analysis.[4] We first present the results of the perfect foresight model, wherein the optimal land allocation decisions are made based on the values of the crop productivity index in the absence of climate and technology shocks. This deterministic analysis is a useful reference point for further discussion when the uncertainty in food crop yields is introduced. We then present the results of the dynamic stochastic model, where the impact of the intrinsic climate and technology uncertainty is brought into the model optimization stage. Specifically, we generate 1,000 sample paths of optimal global land use under different realizations of the stochastic crop productivity index. The results are presented as the difference between the stochastic path and deterministic reference solution.

---

[4]The model converges to its stationary state around 2150. The differences in land use allocations between 2100 and 2150 are small and therefore not reported.





## 4.1 Optimal Path of Global Land Use under Crop Yield Uncertainty

Figure 3 depicts the optimal allocation of global land use over the next century. The left-hand side of Figure 3 shows the
deterministic paths of different types of land considered in this study, i.e., when the food crop yields are perfectly anticipated.
Specifically, it shows three scenarios, where the value of the crop technology index corresponds to (i) expected values of the
stochastic crop productivity index (deterministic-baseline scenario), (ii) the most pessimistic climate and bad technology states
(deterministic-pessimistic scenario), and (iii) the most optimistic climate and good technology states (deterministic-optimistic
scenario). The right-hand side of Figure 3 shows the difference range between the 1,000 simulation paths based on different
ex-ante realizations of the stochastic crop productivity index and the deterministic-baseline path. The 10%, 50% and 90%
quantile lines represent 10%, 50% and 90% quantiles of 1,000 simulated values respectively at each time, and the average line
(the "+" line) represents the average of 1,000 simulated values respectively at each time.

The right-hand side of Figure 3 also shows two extreme cases of optimal land-use paths conditional on period $t$ realizations
of crop productivity index states $A_t(J_{1,t}, J_{2,t})$. The realized crop productivity index always takes the highest possible value
in a *stochastic-optimistic* case (the line of squares), and the lowest possible value in a *stochastic-pessimistic* case (the line
of marks). As future realizations of the stochastic crop productivity index are uncertain, these extreme stochastic solutions
are not the same as the corresponding deterministic solutions, where the values of future crop productivity index are known
with certainty. In the stochastic-optimistic case, for example, potentially lower realizations of future crop productivity index
result in larger current-period agricultural land allocation as compared to the deterministic-optimistic solution. For other model
variables, due to resource limits (e.g., the total land area is unchanged over time) and other constraints, the impact of uncertainty
is theoretically difficult to assess.

We start with the left-hand side of panels (a)-(e) of Figure 3 that shows the optimal land use paths under perfect foresight.
Beginning with the description of the baseline scenario, we see that, in the first half of the coming century, the area dedicated
to food crops increases by 350 million hectares or 22 percent compared to 2004, reaching its maximum of 1.88 billion hectares
around mid-century (panel a). Continuing population growth, intensification of livestock production, and increasing demand
for food, stemming from economic growth are the key drivers for this cropland expansion. In the second half of the coming
century, slower population growth, and technology improvements in crop yields and food processing result in a smaller demand
for cropland. By 2100 cropland area declines significantly relative to its peak value, falling to 1.45 billion hectares, which is,
6 percent smaller than in 2004. In the first half of the coming century land allocation for the second-generation biofuels is
close to zero (panel b). Consistent with the recent analysis of 2G biofuels' deployment potential (National Research Council,
2011), absent aggressive GHG regulations and biofuels' policies, this technology is suboptimal because of the low extraction
and refining costs of fossil fuels and high production and deployment costs of the second-generation biofuels. In the second
half of the coming century, the second-generation technology becomes viable as fossil fuels become scarce and the costs of
producing the second-generation biofuels decline. This results in greater land requirements for the second-generation biofuels
crops. By the end of the coming century, agricultural land dedicated to the second-generation biofuels crops adds 500 million
hectares. Consistent with recent trends, global pasture area declines throughout the entire century (panel c), reflecting increased




a)

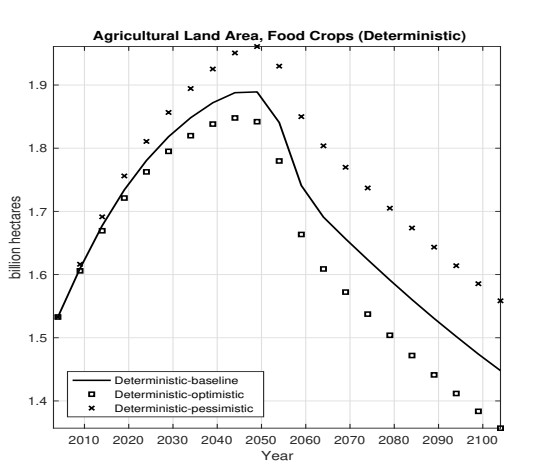
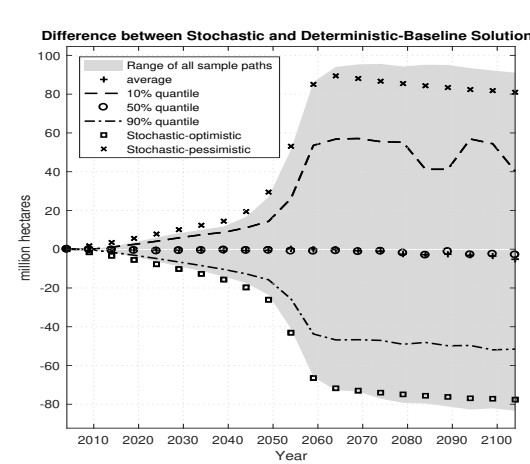

b)

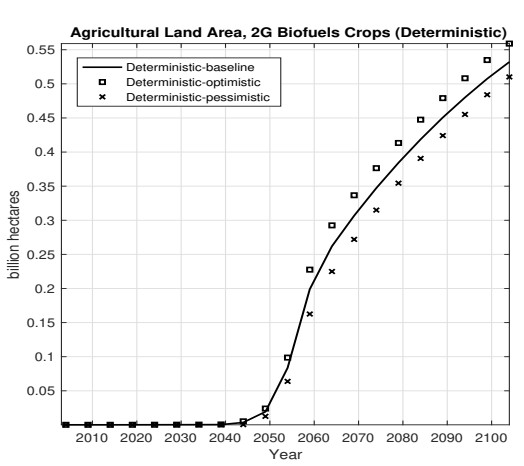
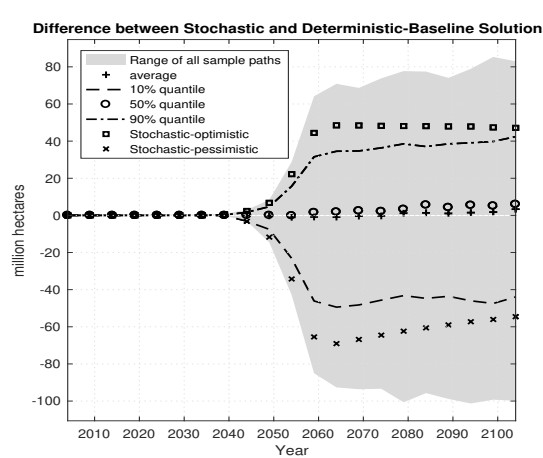

c)

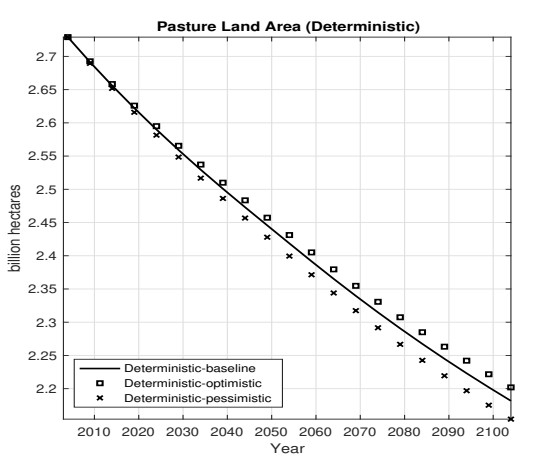
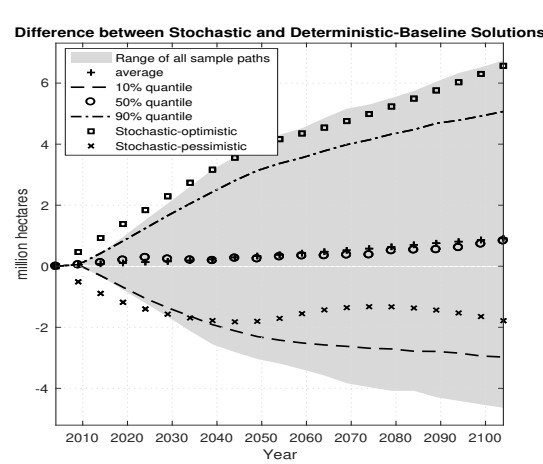



d)

e)

f)

**Figure 3.** Optimal Global Land Use Paths




substitution of pasture land for animal feed in livestock production (Taheripour et al., 2013). Managed forest area increases throughout the entire century, reaching 1.95 billion Ha by 2100 (panel d). In contrast, in response to greater requirements for agricultural land, unmanaged forest area declines by about 500 million Ha over the course of the 21st century (panel e). The decline in unmanaged forest land is less environmentally damaging in the second half of the coming century, as deforestation is limited, with most of the unmanaged forests being converted to managed or protected forest land. Finally, protected forest area more than doubles by the end of the coming century, in light of strong growth in the demand for ecosystem services (panel f). The other two scenarios exhibit broadly similar dynamics. As expected, compared to the baseline scenario, the most pessimistic scenario foresees a greater expansion of agricultural land for food crops and reduction in other types of land (except for protected lands) in response to expected lower realizations of the crop technology index. The situation is reversed for the optimistic scenario. There is a significant variation in the range of anticipated expansion of the agricultural area for food crops between optimistic and pessimistic scenarios, which amounts to 200 million hectares or 14 percent of total cropland in 2100. Variation in expected land use change for other types of land accounts for an approximately equal proportion of the variation in agricultural land for food crops, whereas protected forest areas change very little across different climate scenarios.

Uncertainty in the crop productivity index results in additional redistribution of land resources so to offset the impact of potentially lower yields. As social preferences exhibit intertemporal substitution in this stochastic application of the FABLE model, some of that redistribution takes place even in the absence of the actual changes in the states of climate or technology. Compared to the deterministic scenario, the median (i.e., the 50 percent quantile) path of global land use that corresponds to the "medium" state of climate ($J_{1,t} = 3$) and the "medium" technological state ($J_{2,t} = 2$) foresees a smaller use of agricultural land for food crops (panel a), and greater use of agricultural land for 2G, biofuels' crops (panel b), pasture land (panel c), managed forest land (panel d). There is also a decline in the protected land area (panel e) and an increase in unmanaged natural land (panel d). Unlike the deterministic scenario, almost all of the variation in global land resources in stochastic simulations happens across agricultural land for food crops and 2G biofuels crops (panels a and b). For all other land resources, the differences between stochastic and deterministic paths are small and economically insignificant. This is because land conversion costs of agricultural land for other types of land become larger in the presence of uncertainty. These other types of land have higher adjustment costs of conversion associated with additional time cost of regrowing lumber and livestock, and irreversibilities in accessing protected land areas. Land rotation between food crops and 2G biofuels crops is less costly in the FABLE model. This result is consistent with earlier studies that find that closer integration with the energy sector offers greater potential for food-energy substitution, and thus also a greater resilience against adverse climate conditions affecting food crop yields (Diffenbaugh et al., 2012; Verma et al., 2014).

While the direction of the effect of the uncertainty in the crop productivity on land conversion can be inferred from the economic theory of environmental and natural resource management under uncertainty (see, e.g., Tsur and Zemel (2014) and references therein), the extent to which this uncertainty propagates into land conversion depends critically on chosen model structure and parameters. For example, Alexander et al. (2017, p.1) find that even in the absence of intrinsic uncertainty "systematic differences in land cover areas are associated with the characteristics modeling approach are at least as great as the differences attributed to scenario variations". Depending on the assumptions on the substitution of land for other resources,





the size of technological progress, and the responsiveness of demand for land-based goods and services to changes in crop productivity, this magnitude can be substantially different from other land use models. However, for the same model parameters, we can see that the range of land conversion is considerably smaller for the dynamic stochastic model as compared to the deterministic scenario analysis. As we see from Figure 3, panel (a), the difference between the most extreme paths of the stochastic crop productivity index is about 170 million hectares by 2100, or about 12 percent of the total agricultural area dedicated to food crops. About half of that variation can be attributed to the most extreme (i.e., falling beyond $10^{th}$ and above $90^{th}$ percent quantiles) realizations of crop productivity. This is because the stochastic model assumes that climate and technological states affecting crop yields are reversible (that is if the current state is "bad" (or "good"), it could be "good" (or "bad") in the future). In comparison with the deterministic model under the pessimistic (or optimistic) scenario, the social optimum in the stochastic model requires smaller (or greater) conversion of other types of land to cropland. Thus, agricultural land area in the deterministic pessimistic (or optimistic) scenario is larger (or smaller) than the largest (or the smallest) path in the stochastic simulations. For example, in 2100, under the deterministic-pessimistic scenario, the cropland deviation from the deterministic-baseline scenario is about 100 million hectares, which is 25 percent larger than the largest deviation under the stochastic simulations, and about twice as large as the deviation above the 90% quantile of the stochastic crop technology index. This result demonstrates that scenario analysis can significantly overstate the magnitude of expected agricultural land conversion under uncertain crop yields.

## 4.2 Optimal Path of Land-Based Goods and Services under Crop Yield Uncertainty

The left-hand side panels of Figure 4 report the optimal paths of land-based goods and services under the deterministic model scenarios. Beginning with the baseline scenario, the first panel of Figure 4 shows the production path of food crops, which increases steadily in the first half of the coming century. Compared to 2004, production of food crops (including livestock and biofuels feedstock) nearly doubles, reaching its maximum of about 11 billion tons around 2050. As with cropland expansion, rapid population growth and rising incomes are the key drivers for growing consumption on the demand side. On the supply side, the increase in the production of food crops is further boosted by growing crop yields. At the end of the coming century, production of food crops moderates, as consumers satiate their food requirements and the technology of food marketing and processing improves. By 2100 crop production for the livestock feed has leveled off and even begins to decline. There is also a significant variation in the range of production of food crops between the most optimistic and pessimistic scenarios, which amounts to the sizable amount of 5.3 billion tons.

The results of the dynamic stochastic model simulations also show that uncertainty in the crop productivity index has a profound effect on the optimal production path of food crops. Between the most extreme paths of the stochastic crop productivity index, the production of food crops varies by about 5.4 billion tons compared to the corresponding deterministic path of the stochastic crop productivity index. This is a sizable change, which suggests a significant variation in levels of consumption in 2100 along different paths of the stochastic crop productivity index. In the FABLE model, much of the variation in the optimal path of food crops comes on the demand side, with the crop productivity decline resulting primarily in the reduced consumption of processed crops and livestock. As shown above, the uncertainty-induced supply response is relatively small along the



a)

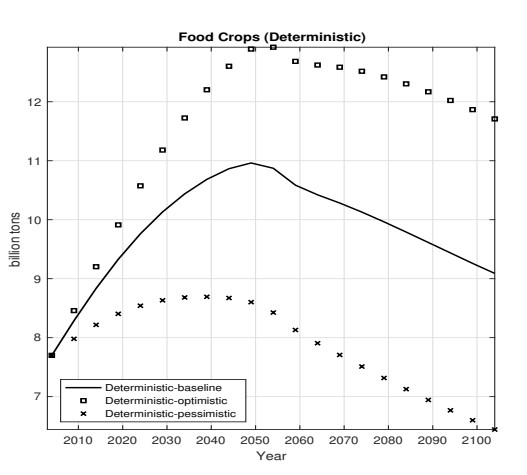

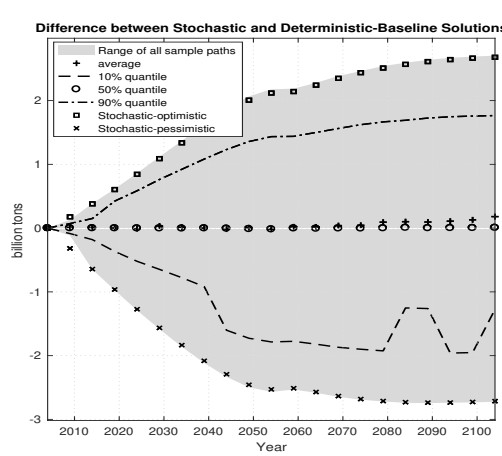

b)

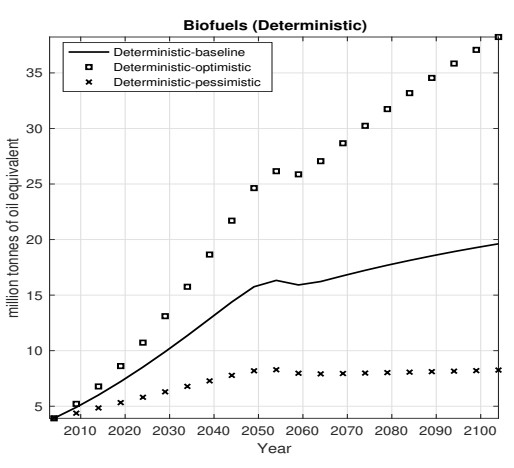

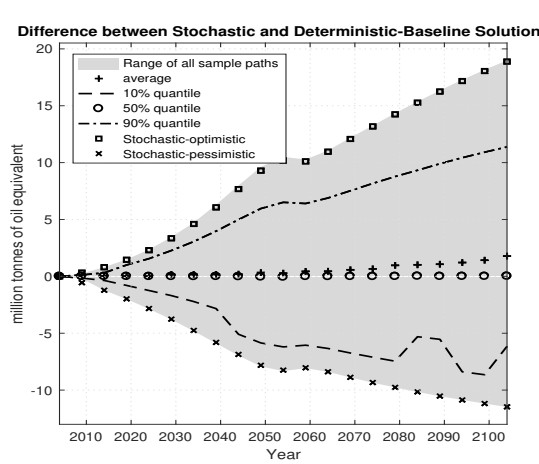

c)

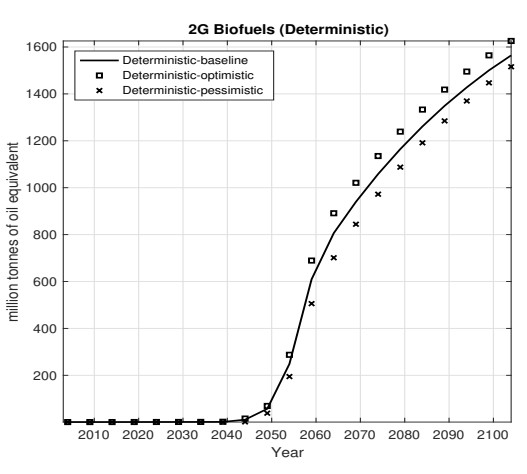

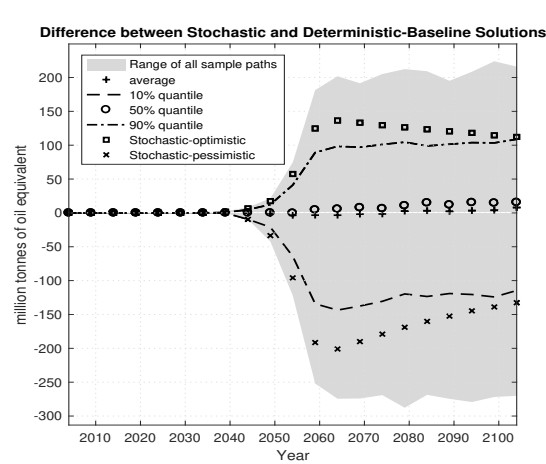



d)



e)

f)

**Figure 4.** Optimal Paths of Land-Based Goods and Services





extensive margin in the dynamic stochastic model (i.e., land conversion). In the appendix (Figure A.2, panel a) we show that the supply response on the intensive margin is smaller, with the ratio of fertilizers to cropland increasing by less than 6 kg/Ha (or eight percent) under extreme realizations of climate and technology uncertainties. About half of that difference, however, corresponds to the most extreme (i.e., falling beyond 10[th] and above 90[th] percent quantiles) realizations of crop productivity.

This result indicates that extreme uncertainty in crop productivity could have a significant impact on food consumption over the coming century.

Production of both first- and second- generation biofuels in the deterministic model grows as oil becomes more scarce along the baseline path and agricultural yields increase (panels b and c). Along that optimal path, characterized by the absence of climate and renewables policies and the abundance of cheap fossil fuels in the first part of the coming century, first-generation

biofuels never become a large source of energy consumption. In 2100 production of first- generation biofuels is a mere 20 million tonnes of oil equivalent (Mtoe) in the baseline scenario and 40 Mtoe in the optimistic scenario. These numbers are considerably higher compared to 2004 but are still small in relative terms (less than one percent of total liquid fuel consumption: see appendix, Figure A.2, panel b). In contrast, the production of second-generation biofuels takes off sharply and expands rapidly after 2040 as they become cost-competitive relative to increasingly costly fossil fuels. In 2100 production of second-

generation biofuels reaches 1.55 billion tonnes of oil equivalent (Btoe) in the baseline scenario. Uncertainty in food crop yields has important implications for the production of the first-generation biofuels that are directly affected by both climate and technology states of food crop yields. The difference between the best and worst states of the crop productivity index is about 31 million tons of oil equivalent, which exceeds their expected baseline production in 2100. Although climate and technology states of food crop yields do not directly affect yields of the second- generation biofuels crops, production of second-generation

biofuels is nonetheless affected through indirect substitution effects of food for energy in the FABLE demand system. There is a sizable variation in the production of second-generation biofuels between extreme paths of the stochastic crop productivity index, which accounts for 450 Mtoe, or about 30 percent of their total production in 2100.

Production of livestock in the deterministic model increases throughout the coming century (panel d), reflecting shifting diets and the growing demand for processed meat as population income increases (Foley et al., 2011). By the end of the coming

century, the production of livestock in the baseline scenario increases by about 1.5 times compared to 2004, reaching 1.28 billion tons. Given the important contribution of livestock feed in the production of livestock, we can see its production is smaller in the pessimistic scenario and larger in the optimistic scenario. The difference in livestock production between the optimistic and pessimistic scenarios accounts for about 550 million tons. This range is similar to the dynamic stochastic model. As the significance of animal feed in livestock production grows over time, the effect of uncertain crop yields becomes more

pronounced. Similar to the result for food crops, the most extreme paths of crop productivity account for about a third of all variation in livestock production.

Production of timber in the deterministic model also expands with the growing demand for timber products and further improvements in forest yields (panel e). By 2100, the production of merchantable timber crops reaches 3.2 billion tons in the deterministic baseline scenario, which is twice as large as in 2004. The consumption of ecosystem services declines in the near

decades, as unmanaged natural forest lands are converted to croplands (panel f). It then increases throughout the remaining part



of the coming century as the demand for ecosystem services increases, and more natural forest lands become institutionally protected. By 2100 consumption of ecosystem services is 36 percent larger than in 2004. Crop productivity has a very small effect on consumption paths of merchantable timber and ecosystem services in either deterministic or stochastic models. This result is not very surprising as crop productivity does not directly affect the production of either timber or ecosystem services, whereas indirect land use change effects are relatively small in this stochastic application of the FABLE model.

## 5 Conclusions

This paper shows the effects of uncertainties associated with nonstationary biophysical processes and technological change on the optimal allocation of natural resources in the long run. In doing so, it applies SCEQ, a cutting-edge computational method for solving nonstationary dynamic high-dimensional stochastic problems to FABLE, a multi-sectoral dynamic model of global land use.

The study focuses on uncertainty in future crop yields, one of the core uncertainties affecting the evolution of global land use in the long run. Combining scenarios from global climate models and high-resolution output from spatial crop simulation models for four major crops, it comes up with a plausible range of realizations of climate shocks and their effect on future crop yields. These estimates are supplemented with an extensive survey of recent agro-economic and biophysical studies assessing the potential for closing yield gaps as well as attaining further advances in potential yields through plant breeding.

The paper's key insight is to illustrate the magnitude of optimal land conversion decisions in the context of different realizations of the stochastic crop productivity. Consistent with the economic theory of natural resource management under uncertainty, the agricultural productivity shocks due either to adverse climate impacts or unexpected limits on further technological progress, resulting in additional conversion of scarce land resources to offset the impact of potentially lower yields. Owing to intertemporal substitution, some of that conversion takes place even in the absence of actual realization of the climate shocks or technology outcomes. This expansion is accompanied by the changes in the consumption of processed food, livestock, and biofuels - the land-based products most affected by changes in crop productivity.

The chosen model (FABLE) seeks to balance computational complexity and economic tractability. It thus ignores many features that are standard in more advanced computational land and other resource use models. Future research should focus on integrating economic decisions under uncertainty into large dynamic natural resource models that feature spatial disaggregation at the regional or zonal level, a more extensive representation of the energy sector and different types of resources and their production derivatives. Another promising research direction would be to incorporate a more detailed representation of uncertain states backed by an econometric analysis that recovers underlying distributions of uncertain natural resource drivers over time.

*Code and data availability.* Numerical implementation of the FABLE model and the SCEQ method in the GAMS modeling language are attached as supplementary material to this manuscript. The FABLE model is calibrated based on the GTAP land use database and publicly





available data sources. Calibration details are available in the appendix. The uncertainty of climate impacts on agricultural yields is estimated on the results of Rosenzweig et al. (2014). Calibration details are available in the appendix.

*Author contributions.* Steinbuks and Cai were the lead contributors to the manuscript. Steinbuks and Hertel developed and calibrated the
FABLE model. Cai developed the SCEQ algorithm and numerically implemented it to quantitatively assess uncertainty in climate and technology impacts. Jaegermeyr and Steinbuks estimated the uncertainty of climate impacts on global crop yields.

*Competing interests.* The contact author has declared that neither they nor their co-authors have any competing interests.

*Acknowledgements.* Cai and Hertel appreciate the financial support from United States Department of Agriculture NIFA-AFRI grant 2015-67023-22905; Cai, Steinbuks, and Hertel appreciate the financial support from the National Science Foundation (SES-0951576 and SES-
1463644) under the auspices of the RDCEP project at the University of Chicago. Cai would also like to thank Becker Friedman Institute at the University of Chicago and Hoover Institution at Stanford University for their support. This paper originates from Cai et al. (2020), which has been separated into this paper and another methodological paper describing the SCEQ computational algorithm.

## Appendix A: FABLE Model Description

This section describes key elements of FABLE model, as well as its equations, variables, and model parameters. For a full
description of the model, including details on model baseline calibration and extensive sensitivity analysis, please refer to Hertel et al. (2016) and Steinbuks and Hertel (2016) and technical appendices therein.

### A1 Primary Resources

Primary resources comprise of land, liquid fossil fuels, and other primary inputs, e.g., labor and capital. The supply of land is fixed and faces competing uses that are determined endogenously by the model. The flow of liquid fossil fuels evolves endoge-
nously along their optimal extraction path, accounting for exogenous discoveries in new fossil fuel reserves. The endowment of other primary inputs is exogenous and evolves along the prespecified global economy growth path.

### A1.1 Land

The total land endowment in the model, $L^{total}$, is fixed. Each period of time $t$ there are four profiles of land in the economy. They include unmanaged forest land, $L^N$, agricultural land, $L^A$, pasture land, $L^P$, and commercially managed forest land, $L^C$.
The agricultural land area can be allocated for the cultivation of food crops (denoted $L^{A,c}$), and second-generation biofuels feedstocks (denoted $L^{A,b2}$). We assume that the natural forest land consists of two types. Institutionally protected land, $L^R$, includes natural parks, biodiversity reserves and other types of protected forests. This land is used to produce ecosystem





services for society, and cannot be converted to commercial land. Unmanaged natural land, $L^N$, can be accessed and either converted to managed land or to protected natural land. Once the natural land is converted to managed land, its potential

to yield ecosystem services is diminished. This potential can be partially restored for managed forests with significant land rehabilitation costs incurred. The use of managed land can be shifted between cropland, forestland, and pasture land (see Figure 1 in the main manuscript for a graphical representation of these transitions). We denote land transition flows from land type $i$ to land type $j$ as $\Delta^{i,j}$ (a negative value means a transition from land type $j$ to land type $i$). Equations describing allocation of land across time and different uses are as follows:

$$L^{total} = \sum_{i=A,P,C,N,R} L_t^i \tag{A1}$$

$$L^A = L^{A,c} + L^{A,b2} \tag{A2}$$

$$L_{t+1}^N = L_t^N - \Delta_t^{N,A} - \Delta_t^{N,R} + \Delta_t^{C,N} \tag{A3}$$


$$L_{t+1}^A = L_t^A + \Delta_t^{N,A} - \Delta_t^{A,P} + \Delta_t^{C,A} \tag{A4}$$

$$L_{t+1}^P = L_t^P + \Delta_t^{A,P} \tag{A5}$$

$$L_{t+1}^R = L_t^R + \Delta_t^{N,R} \tag{A6}$$

Equations (A1) and (A2) define, respectively, the composition of total land and agricultural land in the economy. Equations (A3)-(A5) describe the transitions for unmanaged land, agricultural land, and pasture land.[5] Equation (A6) shows the growth path of protected natural land.

Accessing the natural lands comes at a cost associated with building roads and other infrastructure (Golub et al., 2009).

In addition, converting natural land to reserved land entails additional costs associated with passing legislation to create new natural parks. We denote the natural land access, rehabilitation, and protection costs as $C^{N,A,R}$, $C^{C,N}$, and $C^{N,R}$, respectively. There are also costs of switching between the cropland and the pasture land, denoted as $C^{A,P}$. We assume that all these costs

---

[5]Equations (A2) and (A4) do not account for the transition from forestry to pasture land. Throughout the past century tropical forests, particularly in the Latin America region, have been extensively converted to the pasture land (Barbier et al., 1994). However, in the FABLE model, conversion of forest land to pasture is never optimal as cropland has higher productivity for cattle breeding at the same conversion (stumpage) cost.



are continuous, monotonically increasing, and strictly convex functions of converted land. There are no additional costs of natural land conversion to commercial land, as the revenues from deforestation offset these costs.

Managed forests are characterized by $v_{\max}$ vintages of tree species with vintage ages $v = 1, ..., v_{\max}$. At the end of period $t$ each hectare of managed forest land, $L_{v,t}^C$, has an average density of tree vintage age $v$, with the initial allocation given and denoted by $L_{v,0}^C$. The forest rotation ages and management are endogenously determined. Each period the managed forest land can be either planted, harvested, or left to mature. The newly planted trees occupy $\Delta^{C,C}$ hectares of land, and reach the average age of the first tree vintage next period. The harvested area of tree vintage age $v$ occupies $\Delta_v^{C,H}$ hectares of forest land. The

difference between the harvested area of all tree vintage ages and the newly planted area is used for cropland, i.e.,

$$\Delta_t^{C,A} = \sum_v \Delta_{v,t}^{C,H} - \Delta_t^{C,C}$$

The following equations describe land use of managed forests:

$$L_t^C = \sum_{v=1}^{v_{\max}} L_{v,t}^C, \tag{A7}$$

$$L_{v+1,t+1}^C = L_{v,t}^C - \Delta_{v,t}^{C,H}, \quad v < v_{\max} - 1 \tag{A8}$$

$$L_{v_{\max},t+1}^C = L_{v_{\max},t}^C - \Delta_{v_{\max},t}^{C,H} - \Delta_t^{C,N} + L_{v_{\max}-1,t}^C - \Delta_{v_{\max}-1,t}^{C,H} \tag{A9}$$

$$L_{1,t+1}^C = \Delta_t^{C,C}. \tag{A10}$$

Equation (A7) describes the composition of managed forest area across vintages. Equation (A8) illustrates the harvesting dynamics of forest areas with the ages $v_{\max} - 1$ and $v_{\max}$. Equation (A10) shows the transition from the planted area to new

forest vintage area.

    The average harvesting and planting costs per hectare of new forest planted, $c^{o,H}$, and $c^{o,C}$, are invariant to scale and are the same across all vintages. Harvesting managed forests and conversion of harvested forest land to agricultural land is subject to additional near term adjustment costs, $c^H$. The specific functional forms of land conversion costs are shown in section D, equations (D33)-(D38).

Thus, we have defined the vector of land state variables:

$$\mathbf{L} = \left( L^N, L^A, L^P, L^R, L_1^C, ..., L_{v_{\max}}^C \right)$$

and its associated transition laws.

**A1.2   Fossil Fuels**

The initial stock of liquid fossil fuels, $X^F$, is exogenous, and each period of time $t$ adds a new amount of fossil fuels, $\Delta^{F,D}$,

which reflects exogenous technological progress in fossil fuel exploration. This technological progress comprises of both discoveries on new exploitable oil and gas fields, as well as development of new technologies for extraction of non-conventional fossil fuels. The economy extracts fossil fuels, which have two competing uses in our partial equilibrium model of land-use.





A part of extracted fossil fuels, $\Delta_t^{F,n}$, is converted to fertilizers that are further used in the agricultural sector. The remaining amount of fossil fuels, $\Delta_t^{F,E}$, is combusted to satisfy the demand for energy services. The following equation describes supply of fossil fuels:

$$X_{t+1}^F = X_t^F - \Delta_t^{F,E} - \Delta_t^{F,n} + \Delta_t^{F,D}. \tag{A11}$$

The cost of fossil fuels, $c^F$, reflects the expenditures on fossil fuels' extraction, refining, transportation and distribution, as well the costs associated with emissions control (e.g., Pigovian taxes) in the non-land-based economy. We assume that the cost of fossil fuels is a nonlinear quadratic function with accelerating costs as the stock of fossil fuels depletes (Nordhaus and Boyer, 2000):

$$c_t^F = \xi_1^F \left(\Delta_t^{F,E} + \Delta_t^{F,n}\right)^2 \left(\frac{X_0^F + \Delta_t^{F,D}}{X_t^F + \Delta_t^{F,D}}\right), \tag{A12}$$

where the parameter $\xi_1^F$ captures the curvature of the liquid fossil fuel cost function.

### A1.3 Other Primary Resources

The initial endowment of all other primary resources in the non-land-based economy, such as labor, physical and human capital, and materials inputs, $X^O$, is exogenous in this model. We assume that the growth rate of all other primary resources is a weighted average of the population growth, which reflects demographic changes, and the physical capital growth, $\kappa^{o,X}$. The following equation describes the supply of other primary inputs:

$$X_t^O = X_0^O \left[\alpha^{o,l} \frac{\Pi_t}{\Pi_0} + \left(1 - \alpha^{o,l}\right)\left(1 + \kappa^{o,X}\right)^t\right], \tag{A13}$$

where $\Pi_t$ is the economy's population, and $\alpha^{o,l}$ is the share of population growth to the growth rate of all other primary resources. Other primary inputs can be used for the production of land-based goods and services or converted to final goods and services in the non-land economy. Thus, state variables for resources other than land are defined as:

$$\mathbf{X} = (X^F, X^O).$$

As $X^O$ is exogenous and deterministic, it is a degenerated state variable and not counted as a state variable for model solution purposes.

### A2 Intermediate Inputs

We analyze six intermediate inputs used in the production of land-based goods and services: petroleum products, fertilizers, crops, biofuels, and raw timber. Fossil fuels are refined and converted to either petroleum products, $x^p$, that are further combusted, or to fertilizers, $x^n$, that are used to boost yields in the agricultural sector. Agricultural land and fertilizers are combined to grow food crops, $x^c$ or 2G biofuels crops, $x^{c,b2}$. Food crops can be further converted into processed food and 1G biofuels,



$x^{b1}$, or used as an animal feed, $x^{c,l}$. 2G biofuels crops can only be converted into 2G biofuels, $x^{b2}$. 1G biofuels substitute imperfectly for liquid fossil fuels in final energy demand, whereas 2G biofuels and liquid fossil fuels are the perfect substitutes The food crops used as animal feed and pasture land are combined to produce raw livestock, $x^l$. Harvesting managed forests yield raw timber, $x^w$, that is further used in timber processing. The production functions for intermediate inputs can be illustrated by the following equations

$$x_t^j = g^j \left( \Delta_t^{F,\{E,n\}}, L_t^{\{A,P\}}, \sum_v \Delta_{v,t}^{C,H}, x^{c,\{l,b\}} \right), j = p,n,c,b,l,w. \tag{A14}$$

where $\Delta_t^{F,\{E,n\}}$ represents that either $\Delta_t^{F,E}$ or $\Delta_t^{F,n}$ is an argument of $g^j$, similarly for $L_t^{\{A,P\}}$ and $x^{c,\{l,b\}}$. The specific functional forms of $g^j(\cdot)$ are shown in section D, equations (D13)-(D21).

## A3 Final Goods and Services

We consider five per capita land-based services that are consumed in the final demand: services from processed crops, $y^f$, live-
stock, $y^l$, energy, $y^e$, timber, $y^w$, and ecosystem services, $y^r$. Processed crops, livestock, and timber are respectively products of food crops, raw livestock, and timber processing. The production of energy services combines liquid fossil fuels with the biofuels, and the resulting mix is further combusted. The ecosystem services are the public good to society, which captures recreation, biodiversity, and other environmental goods and services. To close the demand system, we also include other goods and services, $y^o$, which comprise of consumption of other primary inputs not spent on the production of land-based goods and
services. We have defined all state variables for the deterministic model:

$$\mathbf{S} := (\mathbf{L}, \mathbf{X}),$$

and the vector of decision variables

$$\mathbf{a}_t := (\Delta_t^{N,A}, \Delta_t^{N,R}, \Delta_t^{C,N}, \Delta_t^{A,P}, \Delta_t^{C,A}, \Delta_{1,t}^{C,H}, ..., \Delta_{v_{\max},t}^{C,H}, \Delta_t^{C,C}, \Delta_t^{F,E}, \Delta_t^{F,n}, L_t^{A,F}, L_t^{A,B}, \mathbf{x}_t, \mathbf{y}_t),$$

where $\mathbf{x}_t \equiv [x_t^p, x_t^n, x_t^c, x_t^b, x_t^l, x_t^w, x_t^{c,l}, x_t^{c,b}]$ and $\mathbf{y}_t = \left( y_t^f, y_t^l, y_t^e, y_t^w, y_t^r, y_t^o \right)$. The production functions for final per capita
land-based goods and services can be illustrated by the following equation:

$$y_t^i = \mathcal{Y}_t^i (\mathbf{S}_t, \mathbf{a}_t), i = f,l,e,w,r. \tag{A15}$$

where some arguments in $\mathcal{Y}_t^i(\cdot)$ could be redundant. It follows from equation (A15) that production of final goods and services involves the combination of land resources and intermediate inputs. The specific functional forms of $\mathcal{Y}_t^i(\cdot)$ are shown in section D, equations (D22)-(D28), which are functions of $\mathbf{L}$ and $\{x^j\}$. All these equations constitute a part of the feasibility constraint
$\mathbf{a}_t \in \mathcal{D}_t(\mathbf{S}_t)$.

The production of intermediate inputs or final land-based goods and services $i$ incurs costs, $c^{o,i}$, that are subtracted from available other primary resources. The remaining amount of other primary resources is converted into other goods and services,





which are subsequently consumed in final demand. As the focus of this model is on the utilization of land-based resources, we introduce the other goods and services, $y^o$, in a very simplified manner. We introduce no additional cost of producing other
goods and services, assuming that it is reflected in the size of the endowment of other primary inputs. The specific functional form for $y^o$ is shown in section D, equation (D27).

## A4 Preferences

The economy's per-capita utility, $u$, is derived from the per capita consumption of processed crops, livestock, timber, energy and ecosystem services, and other goods and services. Following the macro economic literature, we assume constant relative
risk aversion utility,

$$u(\mathbf{y}) = \frac{\mathcal{C}(\mathbf{y})^{1-\gamma}}{1-\gamma}, \tag{A16}$$

where is the per capita consumption bundle of goods and services, $\mathcal{C}(\mathbf{y})$ is a nonlinear aggregator over $\mathbf{y}$, and $\gamma$ is the coefficient of relative risk aversion, which captures the economy's attitude to uncertain events. We choose a non-homothetic AIDADS preference (Rimmer and Powell, 1996) to compute $\mathcal{C}(\mathbf{y})$ implicitly:

$$\log\left(\mathcal{C}(\mathbf{y})\right) = \sum_{q=f,l,e,w,r,o} \left(\frac{\alpha_q + \beta_q \mathcal{C}(\mathbf{y})}{1 + \mathcal{C}(\mathbf{y})}\right) \log\left(y^q - \underline{y}^q\right) \tag{A17}$$

where $\alpha$, $\beta$, and $\underline{y}^q$ are positive parameters with $\sum_q \alpha_q = \sum_q \beta_q = 1$. These preferences place greater value on eco-system services, and smaller value on additional consumption of food, energy and timber products as society becomes wealthier. When $\gamma = 1$, our utility function is equivalent to the AIDADS utility.

## A5 Welfare

We denote the transition laws of land, (A3)-(A6) and (A8)-(A10), as

$$\mathbf{L}_{t+1} = \mathbf{G}_t^L(\mathbf{S}_t, \mathbf{a}_t), \tag{A18}$$

and the transition laws for other resources, (A11)-(A13), as

$$\mathbf{X}_{t+1} = \mathbf{G}_t^X(\mathbf{S}_t, \mathbf{a}_t). \tag{A19}$$

Combining (A18) and (A19), we have

$$\mathbf{S}_{t+1} = \mathbf{G}_t(\mathbf{S}_t, \mathbf{a}_t) \tag{A20}$$

for the deterministic model in the notations of Section 2.

The objective of the planner is to maximize the total expected welfare, which is the cumulative expected utility of the population's consumption of final goods and services, $\mathbf{y}$, discounted at the constant rate $\delta > 0$. The planner allocates managed





agricultural, pasture and forest lands for crop, livestock, and timber production, the scarce fossil fuels and protected natural forests to solve the following problem:

$$\max_{\mathbf{a}} \sum_{t=0}^{\infty} \delta^t \mathcal{U}(\mathbf{S}_t, \mathbf{a}_t) \tag{A21}$$

subject to the transition laws (A20) and the feasibility constraints

$$\mathbf{F}_t(\mathbf{S}_t, \mathbf{a}_t) \geq 0$$

which include (A15), (D22)-(D28), (D27), (A17) and nonnegativity constraints for the variables. Here

$$\mathcal{U}(\mathbf{S}_t, \mathbf{a}_t) = u(\mathbf{y}_t)\Pi_t$$

is the utility function in the notations of Section 2.

## Appendix B: SCEQ Algorithm

This section briefly presents the SCEQ algorithm (see Cai and Judd (2021) for more details). Following the standard notation in the literature, let $\mathbf{S_t}$ be a vector of state variables (e.g., natural resource stock), and $\mathbf{a}_t$ be a vector of decision variables (e.g., resource extraction, transformation, and final consumption) at each time $t$. The transition law of the state vector $\mathbf{S}$ is

$$\mathbf{S}_{t+1} = \mathbf{G}_t(\mathbf{S}_t, \mathbf{a}_t, \epsilon_t)$$

where $\epsilon_t$ is a serially uncorrelated random vector process, and $\mathbf{G}_t$ is a vector of functions: its $i$-th element, $G_{t,i}$, returns the $i$-th state variable at $t+1$: $S_{t+1,i}$. For simplicity, we assume the mean of $\epsilon_t$ is 0.[6]

We solve the following social planner's problem:

$$\max_{\mathbf{a}_t} \quad \mathbb{E}\left\{ \sum_{t=0}^{T-1} \delta^t \mathcal{U}_t(\mathbf{S}_t, \mathbf{a}_t) + \delta^T V_T(\mathbf{S}_T) \right\} \tag{B1}$$
$$\text{s.t.} \quad \mathbf{S}_{t+1} = \mathbf{G}_t(\mathbf{S}_t, \mathbf{a}_t, \epsilon_t), \ t = 0, 1, 2, ..., T-1,$$
$$\mathbf{F}_t(\mathbf{S}_t, \mathbf{a}_t) \geq 0, \ t = 0, 1, 2, ..., T-1,$$

where $\mathcal{U}_t$ is a utility function, $\delta \in (0,1)$ is the discount factor, $\mathbb{E}$ is the expectation operator, $T$ is the horizon ($T = \infty$ if it is an infinite-horizon problem), $V_T(\mathbf{S}_T)$ is a given terminal value function depending on the terminal state $\mathbf{S}_T$ (it is zero everywhere for an infinite-horizon problem), and $\mathbf{F}_t(\mathbf{S}_t, \mathbf{a}_t) \geq 0$ is a vector of feasibility constraints of actions $\mathbf{a}_t$ at time $t$. And we assume that the initial state $\mathbf{S}_0$ is given, as it can usually be observed or estimated.

---

[6]For notational simplicity we keep the same mathematical representation of a transition function even if some of its elements are redundant. For example, if $G_{t,i}$ is deterministic, we still denote it as $S_{t+1,i} = G_{t,i}(\mathbf{S}_t, \mathbf{a}_t, \epsilon_t)$ even though $S_{t+1,i} = \widetilde{G}_{t,i}(\mathbf{S}_t, \mathbf{a}_t) + 0 \cdot \epsilon_t$. Similarly, if there are some unused elements of $\epsilon_t$ or some redundant arguments in a function $G_{t,j}$, we can multiply them by zero in $G_{t,j}$ and thus still use $S_{t+1,j} = G_{t,j}(\mathbf{S}_t, \mathbf{a}_t, \epsilon_t)$.





In most problems of resource use under uncertainty, the social planner's problem cannot be solved analytically, although certain inferences about potential effects of uncertainty can be made from more stylized models. Numerical dynamic programming with value function iteration (see, e.g., Cai and Judd 2014; Cai 2019) is a typical method to solve these dynamic stochastic problems. However, numerical dynamic programming faces challenging problems such as high dimensionality of state space, shape-preservation of value functions (Cai and Judd, 2013), and kinks caused by occasionally binding constraints. These challenges are common in modeling natural resource use and are hard to address even with the most advanced methods, such as parallel dynamic programming (Cai et al., 2015). For non-stationary problems, value function iteration involves computing decision rules at each period $t$. However, computing all these rules can be very time-consuming and unnecessary if our primary goal is to obtain simulation paths and their distributions until a time of interest, $T^*$, even a long one (in environmental and climate change economics, for example, we are often interested in solutions for the coming century, and set the time of interest to 100 years and the problem horizon of more than 300 years to avoid a large impact of terminal conditions). Instead of solving for optimal decisions for all possible states at each time, we can approximately solve for optimal decisions for those simulated states along simulated paths. The SCEQ algorithm (Cai and Judd, 2021) is for finite or infinite horizon, stationary or nonstationary stochastic dynamic programming problems or competitive equilibrium problems. For simplicity, here we just present its version for finite horizon nonstationary stochastic dynamic programming problems (our model in this study is set to have a 400-year horizon while the time of interest is 100 years).

---

**Algorithm B1** SCEQ for Finite-horizon Stochastic Dynamic Programming Problems with Time-Variant Exogenous Paths

---

**Step 1. Initialization step.** *Given the initial state $\mathbf{S}_0$ and a time of interest $T^*$, as well as a terminal value function $V_T(\mathbf{S}_T)$. Simulate a sequence of $\epsilon_t$ to get $m$ paths, denoted $\epsilon_t^i$ for path $i$, from $t = 0$ to $T^* - 1$. Let $\mathbf{S}_0^i = \mathbf{S}_0$ and iterate forward through steps 2 and 3 for $s = 0, 1, 2, ..., T^* - 1$.*

**Step 2. Optimization step.** *Solve the following deterministic model starting from time $s$ and simulated node $\mathbf{S}_s^i$:*

$$\max_{\mathbf{a}_t} \quad \sum_{t=s}^{T-1} \delta^{t-s} \mathcal{U}_t(\mathbf{S}_t, \mathbf{a}_t) + \delta^{T-s} V_T(\mathbf{S}_T) \tag{B2}$$

$$\text{s.t.} \quad \mathbf{S}_{t+1} = \mathbf{G}_t(\mathbf{S}_t, \mathbf{a}_t, 0), \ t = s, s+1, ..., T-1,$$

$$\mathbf{F}_t(\mathbf{S}_t, \mathbf{a}_t) \geq 0, \ t = s, s+1, ..., T-1,$$

*where $\mathbf{S}_s$ is given by $\mathbf{S}_s^i$, for each $i = 1, ..., m$.*

**Step 3. Simulation step.** *Set $\mathbf{S}_{s+1}^i = \mathbf{G}_t(\mathbf{S}_s^i, \mathbf{a}_s^i, \epsilon_s^i)$, where $\mathbf{a}_s^i$ is the optimal decision at time $s$ of the problem (B2), for each $i = 1, ..., m$.*

---



Algorithm B1 obtains simulated pathways of optimal decisions and states. Note that the inside loop across $i$ can be switched with the outside loop across time, that is, for each $i$, we can obtain one simulation path by iteratively solving (B2) and simulating

$\mathbf{S}_{s+1}^i = \mathbf{G}_t(\mathbf{S}_s^i, \mathbf{a}_s^i, \epsilon_s^i)$ for $s = 0, 1, 2, ..., T^* - 1$.

The optimization step of Algorithm B1 applies the certainty equivalent approximation idea of the NLCEQ method (Cai et al., 2017): for a given state at time $s$, $\mathbf{S}_s^i$, we replace all future stochastic variables by their corresponding expectations conditional on the current state $\mathbf{S}_s^i$,[7] and convert the dynamic stochastic problem (B1) into a deterministic finite-horizon dynamic problem (B2).

We implement the optimal control method (Cai, 2019) to solve (B2) numerically, that is, we view (B2) as a large-scale nonlinear constrained optimization problem with $\{\mathbf{a}_t^i : t \geq s\}$ and $\{\mathbf{S}_t^i : t \geq s\}$ as its variables, and the transition equations and feasibility restrictions as its constraints. The problem can be directly solved with an appropriate nonlinear optimization solver such as CONOPT (Drud, 1994).

Observe that we just need to save the solution of (B2) at time $s$, $\mathbf{a}_s^i$, for use in the next step. In step 3 of Algorithm B1 we use

the saved optimal decision $\mathbf{a}_s^i$ to generate the next-period state, $\mathbf{S}_{s+1}^i = \mathbf{G}_t(\mathbf{S}_s^i, \mathbf{a}_s^i, \epsilon_s^i)$, given realization of shocks, $\epsilon_s^i$. Once we reach the state $\mathbf{S}_{s+1}^i$ at time $s + 1$, we come back to implement step 2 and then step 3. In other words, Algorithm B1 uses an adaptive management way: decisions are made for the current period in the face of the future uncertain shocks; once the next-period shock is observed, decisions for the next period are made with re-optimization given the observed shock and new state variables at the next period. Observe that the serial correlation of random variables has been captured in their associated

transition laws. Repeating this process iteratively through $T^*$ times, we compute a representative simulated pathway of optimal decisions, $\{\mathbf{a}_s^i\}_{s=0}^{T^*-1}$, and states, $\{\mathbf{S}_s^i\}_{s=0}^{T^*}$, which corresponds to the realized path of shocks, $\{\epsilon_s^i\}_{s=0}^{T^*-1}$. Repeating over $i$, we compute $m$ simulated paths of optimal states and decisions and then obtain their distributions. This simulation process can be naturally parallelized.

After we obtain the simulated solutions for our dynamic stochastic land use problem, we also check the normalized Euler

errors and find that the $\mathcal{L}^1$ error of the solutions for the first 100 years (the periods of interest) among 1,000 simulated paths is only $8.6 \times 10^{-4}$, and the corresponding $\mathcal{L}^\infty$ error is only 0.02. This is within range of acceptable accuracy for the most dynamic stochastic problems (Cai and Judd, 2021).

---

[7]As $\epsilon_t$ is a serially uncorrelated stochastic process, we can replace $\epsilon_t$ by its zero mean in the functions of $\mathbf{G}_t$ in (B2) if all transition laws are continuous. For problems with a discrete Markov chain in transition laws, we can use the same technique as described in Cai et al. (2017) for NLCEQ with a discrete stochastic state to obtain the corresponding deterministic model (B1). That is, given realization of the Markov chain at time $s$, we can compute expectations of the Markov chain at all times after $s$ conditional on the value at time $s$ and then replace the stochastic process by the path of the conditional expectations in step 2 of Algorithm B1.





**Appendix C: Quantifying the Uncertainty in Crop Yields**

**C1    Uncertainty in Agricultural Technology**

Advances in crop technology are very difficult to predict due to four interconnected factors (Fischer et al., 2011). First, there is significant uncertainty about the potential for exploiting large and economically significant yield gaps (i.e., the differences between observed and potential crop yields) in developing countries, especially those in Sub-Saharan Africa. A second and closely related point is that it is unclear how fast available yield-enhancing technologies can be adopted at a global scale. Third, there is a significant variation in developing countries' institutions and policies that make markets work better and provide a

conducive environment for agricultural technology adoption. Finally, while plant breeders continue to make steady gains in further advancing crop yields, progress depends on the level of funding provided for agricultural research. This has proven to be somewhat volatile, with per capita funding falling in the decades leading up to the recent food crisis (Alston and Pardey, 2014). The food price rises since 2007 have stimulated new investments. However, whether this interest will be sustained remains to be seen. Overall, progress from conventional breeding is becoming more difficult. Transgenic (genetic modification) technologies

have a proven record of more than a decade of safe and environmentally sound use, and thus offer huge potential to address critical biotic and abiotic stresses in the developing world. However, expected yield gains, costs of further developing these technologies, and the political acceptance of genetically modified foods are all highly uncertain.

To quantify the extent to which the advances in crop technology can further boost agricultural yields over the next century, we first need to assess the magnitude of existing yield gaps at the global scale. In a comprehensive study, Lobell et al. (2009)

report a significant variation in the ratios of actual to potential yields for major food crops across the world, ranging from 0.16 for tropical lowland maize in Sub-Saharan Africa to 0.95 for wheat in Haryana, India. For the purposes of this study, we employ the results of Licker et al. (2010), who conduct comprehensive yield gap analysis using global crop dataset of harvested areas and yields for 175 crops on a 0.5° geographic grid of the planet for the year 2000. Using these estimates, we calculate the global yield gap as the grid-level output-weighted yield gap of the four most important food crops (wheat, maize, soybeans,

and rice). The resulting estimate suggests that average yields are 53% of potential yields, which is close to the median estimates by Lobell et al. (2009). As a further robustness check we employ the Decision Support System for Agrotechnology Transfer (DSSAT) crop simulation model (Jones et al., 2003), run globally on a 0.5 degree grid in the parallel System for Integrating Impacts Models and Sectors (pSIMS; Elliott et al. 2014b) to simulate yields of the same four major food crops under best agricultural management conditions and compare simulated yields to their observed yields. The resulting yield gap estimates

were not substantially different.

In the optimistic (i.e., "good") state of advances in crop technology, we assume that yields continue to grow linearly throughout the coming century, eliminating the yield gap by 2100. This high yield scenario rests on the assumption of continued strong growth in investment in agricultural research and development, widespread acceptance of genetically modified crops, continuing institutional reforms in developing countries, and public and private investments in the dissemination of new technologies.

The erosion of any one of these component assumptions will likely result in a slowing of crop technology improvements. And



there are some grounds for pessimism. In a comprehensive statistical analysis of historical crop production trends, Grassini et al. (2013) note that

> "despite the increase in investment in agricultural R&D and education [...] the relative rate of yield gain for the major food crops has decreased over time together with evidence of upper yield plateaus in some of the most
> productive domains. For example, investment in R&D in agriculture in China has increased threefold from 1981 to 2000. However, rates of increase in crop yields in China have remained constant in wheat, decreased by 64% in maize as a relative rate and are negligible in rice. Likewise, despite a 58% increase in investment in agricultural R&D in the United States from 1981 to 2000 (sum of public and private sectors), the rate of maize yield gain has remained strongly linear."

To capture the possibility of much slower technological improvement in the coming century, we specify two more pessimistic scenarios. In the "medium" state of technology, rather than closing the yield gap by 2100, average yields in 2100 are just three-quarters of yield potential at that point in time. In the "bad" state of technology, there is no technological progress, and the crop yields stay the same as at the beginning of the coming century. This is the path on which we begin the simulation in 2004. As previously noted, we then specify probabilities with which the crop technology index evolves across the different states of technology.

## C2 Uncertainty in Climate Change Impacts

In addition to crop technology uncertainty, there is great uncertainty about the physical environment in which this technology will be deployed. In particular, long-run changes in both temperature and precipitation are likely to have an important impact on the productivity of land in agriculture (IPCC, 2014a), and therefore, the global pattern of land use. Quantification of the impact of climate change on agricultural yields requires coming to grips with three interconnected factors (Alexandratos, 2011). First, there is significant uncertainty in future GHG concentrations along the long-run growth path of the global economy. Second, the General Circulation Models (GCMs) developed by climate scientists to translate these uncertain GHG concentrations into climate outcomes disagree about the spatially disaggregated deviations of temperature and precipitation from baseline levels. Finally, there is significant uncertainty in the biophysical models used to determine how changes in temperature and precipitation will affect plant growth and the productivity of agriculture in different agro-ecological conditions. The impact of climate change on food crop yields depends critically on their phenological development, which, in turn, depends on the accumulation of heat units, typically measured as growing degree days (GDDs). More rapid accumulation of GDDs as a result of the climate change speeds up phenological development, thereby shortening key growth stages, such as the grain filling stage, hence reducing potential yields (Long, 1991). However, rising concentrations of $CO_2$ in the atmosphere result in an increase in potential yields due to improved water use efficiency, often dubbed the "$CO_2$ fertilization effect" (Long et al., 2006). Sorting out the relative importance of these effects and achieving greater confidence in evaluations of climate impacts on agricultural yields remains an important research question in the agronomic literature (Cassman et al., 2010; Rosenzweig et al., 2014).



To quantify the uncertainty in climate impacts on agricultural yields we follow the approach of Rosenzweig et al. (2014), who have recently conducted a globally consistent, protocol-based, multi-model climate change assessment for major crops with the explicit characterization of uncertainty. To quantify the uncertainty of impacts of temperature increases due to climate change on potential crop yields we obtain results of four crop simulation models: GEPIC (Liu et al., 2007), LPJmL (Bondeau et al., 2007), pDSSAT (Jones et al., 2003), and PEGASUS (Deryng et al., 2011). All models are run globally on a 0.5° grid over the period between 1971 and 2099 and weighted by the agricultural output of four major food crops (maize, soybeans, wheat, and rice). To ensure simulation results comparability with the structural parameters of FABLE model all models are run under Representative Concentration Pathways $6.0\mathrm{W/m^2}$ (RCP6) GHG forcing scenario (Moss et al., 2008). We also consider alternative assumptions on $CO_2$ fertilization effects. Observe that our results are based on four crop simulation models though Rosenzweig et al. (2014) consider seven crop simulation models. The remaining three models have fewer crops and/or temporal frames for model baseline and are thus omitted. Rosenzweig et al. (2014) find that five models, including GEPIC, LPJmL, and pDSSAT models considered in this analysis, yield broadly similar predictions. One model (LPJ-GUESS) not covered here has much higher variation in predicted crop yields under different climate scenarios. Our results may, therefore, understate the range of uncertainty of climate change impacts on potential crop yields.

To quantify uncertainty in temperature increases due to climate change we employ outputs for five global climate models (GCM): GFDL-ESM2M (Dunne et al., 2013), HadGEM2-ES (Collins et al., 2008), IPSL-CM5A-LR (Dufresne et al., 2012), MIROC-ESM-CHEM (Watanabe et al., 2011), and NorESM1-M (Bentsen et al., 2012). For each of the simulations, we fit a linear trend in order to parsimoniously characterize the evolution of crop yields in the face of climate change over the coming century.

Figure A1 summarizes simulation results for four crop simulation models and five climate models (with and without fertilization effects) in 2100, normalized relative to assumed yield potential in the absence of climate change. There is significant heterogeneity in terms of both direction and magnitude of climate impacts on agricultural yields across global climate models when the $CO_2$ fertilization effect is considered.[8] Regardless of the chosen climate model, for the scenario with fertilization effects, two out of four crop simulation models (LPJmL and pDSSAT) predict a moderate increase in potential yields (5-15 percent), whereas the PEGASUS model predicts a large decline in potential yields (20-30 percent). The GEPIC model predicts that on average crop yields will be little changed, showing a small increase in crop yields for some climate models and a small decline for other models. The predictions of LPJmL and pDSSAT models are reversed when $CO_2$ fertilization effects are removed, showing a decline of about 10-15 percent in potential yields. The PEGASUS model predicts an even larger decline in potential yields (30-35 percent), whereas the predictions of GEPIC model show a moderate decline of about 5-10 percent in potential yields.

---

[8]Field trials show that higher atmospheric $CO_2$ concentrations enhance photosynthesis and reduce crop water stress (Deryng et al., 2016). This fertilization effect interacts with other factors such as nutrient availability, and current-generation crop models are characterized by large uncertainties regarding net $CO_2$ fertilization potentials at larger spatial scales. In line with previous studies (Rosenzweig et al., 2014; Elliott et al., 2014a; Jägermeyr et al., 2016) we use a constant $CO_2$ case as pessimistic assumption regarding climate change effects, and a transient $CO_2$ case according to the RCP concentration pathways to reflect a more optimistic case.





Given a large variation in model predictions, we construct 5 states for potential crop yields under uncertain climate change.

These states correspond to quintiles of the distribution of different model outcomes for potential crop yields by 2100. Under two optimistic states of the world, we observe a 2 and 15 percent increases in potential crop yields relative to model baseline whereby significant $CO_2$ fertilization effects offset the negative effects of climate change. For the next two states, we see a 15 and 19 percent declines in potential crop yields relative to model baseline whereby $CO_2$ fertilization effects are either small or nonexistent, and the negative effects of climate change tend to prevail. Finally, under most pessimistic states of the world,

drastic adverse effects of climate change combined with the absence of any $CO_2$ fertilization effects result in a 36 percent decline in potential crop yields relative to model baseline.

## C3 Transition probabilities

The five possible values of the climate state $J_{1,t}$ are $\mathcal{J}_{1,1} = 0.64$, $\mathcal{J}_{1,2} = 0.85$, $\mathcal{J}_{1,3} = 0.89$, $\mathcal{J}_{1,4} = 1.02$, and $\mathcal{J}_{1,5} = 1.15$, and its probability transition matrix is

$$
\quad P_1 = \begin{bmatrix} 0.5 & 0.25 & & & \\ 0.5 & 0.5 & 0.25 & & \\ & 0.25 & 0.5 & 0.25 & \\ & & 0.25 & 0.5 & 0.5 \\ & & & 0.25 & 0.5 \end{bmatrix}
$$

where $P_{1,i,j}$ represents the probability from the $j$-th value of $J_{1,t}$ to the $i$-th value, for $1 \leq i,j \leq 5$. The three possible values of the technological state $J_{2,t}$ are $\mathcal{J}_{2,1} = 1.45$, $\mathcal{J}_{2,2} = 1.675$, and $\mathcal{J}_{2,3} = 1.9$, and its probability transition matrix is

$$
P_2 = \begin{bmatrix} 0.4423 & 0.1416 & 0.1311 \\ 0.4139 & 0.669 & 0.4367 \\ 0.1438 & 0.1894 & 0.4322 \end{bmatrix},
$$

where $P_{2,i,j}$ represents the probability from the $j$-th value of $J_{2,t}$ to the $i$-th value for $1 \leq i,j \leq 3$. We assume that $J_{2,t}$ is

independent of $J_{1,t}$.

## C4 Model

After we add the risks, the state vector becomes

$$\mathbf{S} := (\mathbf{L}, \mathbf{X}, \mathbf{J})$$

where $\mathbf{J}_t = (J_{1,t}, J_{2,t})$. And $\mathbf{J}$ is a Markov chain so it can be represented as $\mathbf{J}_{t+1} = \mathbf{G}_t^J(\mathbf{J}_t, \epsilon_t)$ where $\epsilon_t$ is a vector of shocks

with zero means. The problem is

$$\max_{\mathbf{a}} \mathbb{E}\left\{\sum_{t=0}^{\infty} \delta^t \mathcal{U}(\mathbf{S}_t, \mathbf{a}_t)\right\} \tag{C1}$$





subject to

$$\mathbf{L}_{t+1} = \mathbf{G}_t^L(\mathbf{S}_t, \mathbf{a}_t)$$

$$\mathbf{X}_{t+1} = \mathbf{G}_t^X(\mathbf{S}_t, \mathbf{a}_t)$$

$$\mathbf{J}_{t+1} = \mathbf{G}_t^J(\mathbf{J}_t, \epsilon_t)$$

and $\mathbf{a}_t \in \mathcal{D}_t(\mathbf{S}_t)$ representing the feasibility constraints, that is, inequality constraints and the equations other than the above transition laws. The above transition laws are just a special case of

$$\mathbf{S}_{t+1} = \mathbf{G}_t(\mathbf{S}_t, \mathbf{a}_t, \epsilon_t)$$

in the notations of Section 2 of this Appendix, so we can implement the SCEQ method to solve the dynamic stochastic
programming problem. Since our time of interest is $T^* = 100$ years, we change the problem (C1) to have a finite horizon with $T = 400$ years as a larger $T$ has little impact on our solution in the first 100 years.

In the step 2 of Algorithm 1 for the solution at time $s$, we replace $\epsilon_t$ by its zero mean to have $\mathbf{S}_{t+1} = \mathbf{G}_t(\mathbf{S}_t, \mathbf{a}_t, 0)$, that is, $\mathbf{J}_{t+1} = \mathbf{G}_t^J(\mathbf{J}_t, 0)$. But this $\mathbf{J}_{t+1} = \mathbf{G}_t^J(\mathbf{J}_t, 0)$ is only for simplicity in notations. In fact, since $\mathbf{J}$ is a Markov chain, we replace $\mathbf{J}_t$ by its mean conditional on the realized value of $\mathbf{J}_s$ (i.e., its certainty equivalent approximation):

$$[\mathcal{J}_1 \pi_{1,t,s}, \ \mathcal{J}_2 \pi_{2,t,s}]$$

for all $t \geq s$, where $\mathcal{J}_1 = (\mathcal{J}_{1,1}, ..., \mathcal{J}_{1,5})$, $\mathcal{J}_2 = (\mathcal{J}_{2,1}, \mathcal{J}_{2,2}, \mathcal{J}_{2,3})$, $\pi_{1,t,s}$ and $\pi_{2,t,s}$ are two column vectors representing probability distributions of $J_{1,t}$ and $J_{2,t}$ conditional on the realized values of $J_{1,s}$ and $J_{2,s}$ respectively. If the realized values of $J_{1,s}$ and $J_{2,s}$ are $\mathcal{J}_{1,i}$ and $\mathcal{J}_{2,j}$ respectively, then we have $\pi_{1,t,s} = P_1^{t-s} \pi_{1,s,s}$ and $\pi_{2,t,s} = P_2^{t-s} \pi_{2,s,s}$, where $\pi_{1,s,s}$ is a length-5 column vector with 1 at the $i$th element and 0 everywhere else, and $\pi_{2,s,s}$ is a length-3 column vector with 1 at the $j$th element
and 0 everywhere else.

## Appendix D: Model Equations, Variables and Parameters

### D1   Equations

*Land Use*

$$L = \sum_{i=A,P,C,N,R} L_t^i \tag{D1}$$


$$L_{t+1}^N = L_t^N - \Delta_t^{N,A} - \Delta_t^{N,R} + \Delta_t^{C,N} \tag{D2}$$

$$L_t^A = L_t^{A,c} + L_t^{A,b2} \tag{D3}$$





$$L_{t+1}^A = L_t^A + \Delta_t^{N,A} - \Delta_t^{A,P} + \Delta_t^{C,A} \tag{D4}$$

$$L_{t+1}^P = L_t^P + \Delta_t^{A,P} \tag{D5}$$

$$L_{t+1}^R = L_t^R + \Delta_t^{N,R} \tag{D6}$$

$$L_t^C = \sum_{v=1}^{v_{\max}} L_{v,t}^C, \tag{D7}$$

$$L_{v+1,t+1}^C = L_{v,t}^C - \Delta_{v,t}^{C,H}, \quad v < v_{\max} - 1 \tag{D8}$$

$$L_{v_{\max},t+1}^C = L_{v_{\max},t}^C - \Delta_{v_{\max},t}^{C,H} - \Delta_t^{C,N} + L_{v_{\max}-1,t}^C - \Delta_{v_{\max}-1,t}^{C,H}, \tag{D9}$$

$$L_{1,t+1}^C = \Delta_t^{C,C} \tag{D10}$$

$$\Delta_{v,t}^{C,H} \leq L_{v,t}^C, \quad v < v_{\max}$$

$$\Delta_{v_{\max},t}^{C,H} + \Delta_t^{C,N} \leq L_{v_{\max},t}^C$$

$$\Delta_t^{C,A} = \sum_{v=1}^{v_{\max}} \Delta_{v,t}^{C,H} - \Delta_t^{C,C}$$

*Fossil Fuels*

$$X_{t+1}^F = X_t^F - \Delta_t^{F,E} - \Delta_t^{F,n} + \Delta_t^{F,D} \tag{D11}$$





*Other Primary Resources*

$$X_t^O = X_0^O \left[ \alpha^{o,l} \frac{\Pi_t}{\Pi_0} + \left(1 - \alpha^{o,l}\right) \left(1 + \kappa^{o,2}\right)^t \right] \tag{D12}$$

*Intermediate Products*

$$x_t^p = \theta_t^p \Delta_t^{F,E} \tag{D13}$$

$$x_t^n = \theta^n \Delta_t^{F,n} \tag{D14}$$

$$x_t^n = x_t^{n,c} + x_t^{n,b2} \tag{D15}$$

$$x_t^c = \theta_t^c \left( \alpha^n \left( L_t^{A,c} \right)^{\rho_n} + \left(1 - \alpha^n\right) \left( x_t^{n,c} \right)^{\rho_n} \right)^{\frac{1}{\rho_n}} \tag{D16}$$

$$x_t^{c,b2} = \theta_t^{c,b2} \left( \alpha^n \left( L_t^{A,b2} \right)^{\rho_n} + \left(1 - \alpha^n\right) \left( x_t^{n,b2} \right)^{\rho_n} \right)^{\frac{1}{\rho_n}} \tag{D17}$$

$$x_t^{b1} = \theta^{b1} x_t^{c,b} \tag{D18}$$

$$x_t^{b2} = \theta^{b2} \left( \left( \alpha^{b2} \right)^{\theta_t^{b2,K}} (K)^{\rho_{b2}} + \left(1 - \alpha^{b2}\right) \left( x_t^{c,b2} \right)^{\rho_{b2}} \right)^{\frac{1}{\rho_{b2}}} \tag{D19}$$

$$x_t^l = \theta^P \left( \alpha^l \left( L_t^P \right)^{\rho_l} + \left(1 - \alpha^l\right) \left( x_t^{c,l} \right)^{\rho_l} \right)^{\frac{1}{\rho_l}} \tag{D20}$$

$$x_t^w = \sum_{v=1}^{v_{\max}} \theta_{v,t}^w \Delta_{v,t}^{C,H} \tag{D21}$$

*Final Goods and Services*

$$Y_t^f = \theta_t^f \left( x^c - x^{c,b} - x^{c,l} \right) \tag{D22}$$





$$Y_t^e = \theta_t^e \left( \alpha^e \left( x_t^{b1} \right)^{\rho_e} + (1 - \alpha^e) \left( x_t^p + x_t^{b2} \right)^{\rho_e} \right)^{\frac{1}{\rho_e}} \tag{D23}$$

$$Y_t^l = \theta_t^l x_t^l, \tag{D24}$$

$$Y_t^w = \theta_t^{y_w} x_t^w \tag{D25}$$

$$Y_t^r = \theta^r \left[ \sum_{i=A,P,C} \alpha^{i,r} \left( L_t^i \right)^{\rho_r} + \left( 1 - \sum_{i=A,P,C} \alpha^{i,r} \right) \left( L_t^N + \theta^R L_t^R \right)^{\rho_r} \right]^{\frac{1}{\rho_r}} \tag{D26}$$

$$Y_t^o = \theta_t^{o,1} \begin{bmatrix} X_t^O - \frac{1}{\theta_0^o}[c^{o,c} \frac{x_t^c}{A_t} + c^{o,cb} \frac{x_t^{c,b2}}{\theta_t^{c,b2}} + c^{o,f} \frac{Y_t^f}{\theta_t^f} + c^{o,p} x_t^p + c^{o,n} x_t^n + c^{o,b} x_t^{b1} \\ + c^{o,b2} x_t^{b2} + c^{o,l} x_t^l + c^{o,yl} \frac{\theta_0^l Y_t^l}{\theta_t^l} + c_t^{o,w} \Delta_t^{C,H} + c^{o,y_w} x_t^w \\ + c^{o,r} L_t^R + c^p \Delta_t^{C,C} + C_t^N + C_t^R + C_t^F + C_t^H + C_t^P + C_t^{C,N}] \end{bmatrix} \tag{D27}$$

$$\mathbf{y}_t = \left( y_t^f, y_t^l, y_t^e, y_t^w, y_t^r, y_t^o \right) = \left( Y_t^f, Y_t^l, Y_t^e, Y_t^w, Y_t^r, Y_t^o \right) / \Pi_t \tag{D28}$$

*Technology (deterministic)*

$$A_t = \frac{A_T A_0 e^{\kappa_c t}}{A_T + A_0 \left( e^{\kappa_c t} - 1 \right)} \tag{D29}$$

$$\theta_{v,t}^w = \begin{cases} 0.00001 \text{ if } v \leq \underline{v} \\ \overline{\theta}_v^w (1 + \kappa_v^w t) \text{ if } v > \underline{v} \end{cases}, \overline{\theta}_v^w = \exp\left( \psi_a - \frac{\psi_b}{(v - \underline{v})} \right) \tag{D30}$$


$$\theta_t^i = \theta_0^i (1 + \kappa^i)^t, i = f, e, l, y^w, o \tag{D31}$$

*Technology (stochastic)*

$$A_t = \frac{A_T(J_{1,t}, J_{2,t}) A_0 e^{\kappa_c t}}{A_T(J_{1,t}, J_{2,t}) + A_0 \left( e^{\kappa_c t} - 1 \right)} \tag{D32}$$





*Costs*

$$C_t^{N,A,R} = \xi_0^n \left( \Delta_t^{N,A} + \Delta_t^{N,R} \right) + \xi_1^n \left( \Delta_t^{N,A} + \Delta_t^{N,R} \right)^2 \tag{D33}$$

$$C_t^{N,R} = \xi_0^R \Delta_t^{N,R} + \xi_1^R \left( \Delta_t^{N,R} \right)^2 \tag{D34}$$

$$C_t^F = \xi_1^F \left( \Delta_t^{F,E} + \Delta_t^{F,n} \right)^2 \left( \frac{X_0^F + \Delta^{F,D}}{X_t^F + \Delta^{F,D}} \right) \tag{D35}$$

$$C_t^H = \xi_0^H \left( \Delta_t^{C,H} - \Delta_t^{C,C} \right)^2 + \sum_v \frac{\xi_1^H}{L_{v,t+1}^C + \xi_2^H} \tag{D36}$$

$$C_t^{A,P} = \xi_1^P \left( \Delta_t^{A,P} \right)^2 \tag{D37}$$

$$C_t^{C,N} = \xi_0^{C,N} \Delta_t^{C,N} + \xi_1^{C,N} \left( \Delta_t^{C,N} \right)^2 \tag{D38}$$

*Preferences*

$$u(\mathbf{y}) = \frac{\mathcal{C}(\mathbf{y})^{1-\gamma}}{1-\gamma} \tag{D39}$$

$$\log \left( \mathcal{C}(\mathbf{y}) \right) = \sum_{q=f,l,e,w,r,o} \left( \frac{\alpha_q + \beta_q \mathcal{C}(\mathbf{y})}{1 + \mathcal{C}(\mathbf{y})} \right) \log \left( y_t^q - \underline{y}^q \right) \tag{D40}$$

*Population*

$$\Pi_t = \frac{\Pi_T \Pi_0 e^{\kappa^\pi t}}{\Pi_T + \Pi_0 \left( e^{\kappa^\pi t} - 1 \right)} \tag{D41}$$

*Welfare*

$$\Omega = \mathbb{E} \left\{ \sum_{t=0}^{\infty} \delta^t \mathcal{U}(\mathbf{S}_t, \mathbf{a}_t) \right\}. \tag{D42}$$

with $\mathcal{U}(\mathbf{S}_t, \mathbf{a}_t) = u(\mathbf{y}_t)\Pi_t$, $\mathbf{S} := (\mathbf{L}, \mathbf{X}, \mathbf{J})$, and

$$\mathbf{a}_t := (\Delta_t^{N,A}, \Delta_t^{N,R}, \Delta_t^{C,N}, \Delta_t^{A,P}, \Delta_t^{C,A}, \Delta_{1,t}^{C,H}, ..., \Delta_{v_{\max},t}^{C,H}, \Delta_t^{C,C}, \Delta_t^{F,E}, \Delta_t^{F,n}, L_t^{A,F}, L_t^{A,B}, \mathbf{x}_t, \mathbf{y}_t).$$



**Figure A1.** Changes in Potential Crop Yields under RCP 6 Scenario in 2100







**Figure A2.** Consumption of Fertilizers and Biofuels



**Table A1.** Model Exogenous Variables

| Parameter | Description | Units |
|---|---|---|
| *Exogenous Variables* | | |
| $\Delta_t^{F,D}$ | Flow of Newly Discovered Fossil Fuels | trillion toe |
| $X_t^O$ | Other Primary Goods | trillion USD |
| $A_t$ | Crop Technology Index | |
| $\theta_t^{c,b2}$ | 2G biofuels Crop Technology Index | |
| $\theta_t^{b2,K}$ | 2G Biofuels Fixed Factor Decay Index | |
| $\theta_{v,t}^{w}$ | Logging Productivity Index | |
| $\theta_t^{f}$ | Food Processing Productivity Index | |
| $\theta_t^{e}$ | Energy Efficiency Index | |
| $\theta_t^{l}$ | Livestock Processing Productivity Index | |
| $\theta_t^{y^w}$ | Wood Processing Productivity Index | |
| $\theta_t^{o}$ | Total Factor Productivity Index | |
| $C_t^F$ | Fossil Fuel Extraction Cost | share of $X_t^O$ |
| $C_t^N$ | Natural Land Access Cost | share of $X_t^O$ |
| $C_t^R$ | Natural Land Protection Cost | share of $X_t^O$ |
| $C_t^H$ | Managed Forest Conversion Cost | share of $X_t^O$ |
| $C_t^P$ | Pasture Land Conversion Cost | share of $X_t^O$ |
| $C_t^{C,N}$ | Natural Land Restoration Cost | share of $X_t^O$ |
| $\Pi_t$ | Population | billion people |





**Table A2.** Model Endogenous Variables

| Parameter | Description | Units |
|---|---|---|
| $L_t^A$ | Agricultural Land Area | GHa |
| $L_t^{A,c}$ | Agricultural Land Area, food crops | GHa |
| $L_t^{A,b2}$ | Agricultural Land Area, 2G biofuels crops | GHa |
| $L_t^P$ | Pasture Land Area | GHa |
| $L_t^C$ | Commercial Forest Land Area | GHa |
| $L_t^N$ | Unmanaged Natural Land Area | GHa |
| $L_t^R$ | Protected Natural Land Area | GHa |
| $\Delta_t^{N,A}$ | Flow of Deforested Natural Land | GHa |
| $\Delta_t^{N,R}$ | Flow of Protected Natural Land | GHa |
| $\Delta_t^{C,N}$ | Flow of Restored Natural Land | GHa |
| $\Delta_t^{C,A}$ | Managed Forest Land Converted to Agriculture | GHa |
| $\Delta_t^{C,C}$ | Replanted Forest Land Area | GHa |
| $\Delta_{v,t}^{C,H}$ | Harvested Forest Land Area of Vintage $v$ | GHa |
| $\Delta_t^{A,P}$ | Agricultural Land Converted to Pasture | GHa |
| $X_t^F$ | Stock of Fossil Fuels | Ttoe |
| $\Delta_t^{F,E}$ | Flow of Fossil Fuels Converted to Petroleum | Ttoe |
| $\Delta_t^{F,n}$ | Flow of Fossil Fuels Converted to Fertilizers | Ttoe |
| $x_t^p$ | Petroleum Products | Gtoe |
| $x_t^n$ | Fertilizers | Gton |
| $x_t^c$ | Food Crops | Gton |
| $x_t^{c,b2}$ | 2G Biofuels Crops | Gton |
| $x_t^{b1}$ | 1G Biofuels | Gtoe |
| $x_t^{b2}$ | 2G Biofuels | Gtoe |
| $x_t^l$ | Livestock | Gtoe |
| $x_t^w$ | Raw Timber | Gton |
| $Y_t^f$ | Services from Processed Food | billion USD |
| $Y_t^e$ | Energy Services | billion USD |
| $Y_t^l$ | Services from Processed Livestock | billion USD |
| $Y_t^w$ | Services from Processed Timber | billion USD |
| $Y_t^r$ | Eco-system Services | billion USD |
| $Y_t^o$ | Other Goods and Services | trillion USD |





**Table A3.** Baseline Parameters

| Parameter | Description | Units | Value |
|---|---|---|---|
| *Population* | | | |
| $\Pi_0$ | Population in 2004 | billion people | 6.39 |
| $\Pi_T$ | Population in time T | billion people | 10.1 |
| $\kappa^\pi$ | Population Convergence Rate | | 0.042 |
| *Land Use* | | | |
| $L$ | Total Land Area | billion Ha | 8.56 |
| $L_0^A$ | Area of Agricultural Land in 2004 | billion Ha | 1.53 |
| $L_0^P$ | Area of Pasture Land in 2004 | billion Ha | 2.73 |
| $L_0^C$ | Area of Commercial Forest Land in 2004 | billion Ha | 1.62 |
| $L_0^N$ | Area of Unmanaged Natural Land in 2004 | billion Ha | 2.47 |
| $L_0^R$ | Area of Protected Natural Land in 2004 | billion Ha | 0.207 |
| $\xi_0^n$ | Access Cost Function Parameter | | 0.6 |
| $\xi_1^n$ | Access Cost Function Parameter | | 105 |
| $\xi_0^R$ | Protection Cost Function Parameter | | 4.5 |
| $\xi_1^R$ | Protection Cost Function Parameter | | 400 |
| $\xi_1^P$ | Pasture Conversion Cost Function Parameter | | 170 |
| $\xi_0^H$ | Forest Conversion Cost Function Parameter | | 80 |
| $\xi_1^H$ | Forest Conversion Cost Function Parameter | | 0.004 |
| $\xi_0^{C,N}$ | Natural Land Restoration Cost Parameter | | 0.8 |
| $\xi_1^{C,N}$ | Natural Land Restoration Cost Parameter | | 400 |
| *Fossil Fuels* | | | |
| $X_0^F$ | Endowment of Fossil fuels in 2004 | trillion toe | 0.343 |
| $\Delta^{F,D}$ | Flow of Newly Discovered Fossil Fuels | trillion toe | 0.008 |
| $\xi_1^F$ | Fuel Extraction Cost Function Parameter | | 2000 |
| *Other Primary Goods* | | | |
| $X_0^O$ | Endowment of Other Primary Goods in 2004 | USD $\times 10^{13}$ | 3.16 |
| $\kappa^{o,X}$ | Growth Rate of Physical Capital | | 0.0035 |
| $\alpha^{o,l}$ | Share of demographic factors in growth of $X_t^O$ | | 0.39 |
| *Intermediate Products* | | | |
| $\theta^p$ | Petroleum Conversion Factor | per toe of $\Delta_t^{F,E}$ | 0.5 |
| $c^{o,p}$ | Petroleum Conversion Cost | share of $X_t^O$ | 0.0157 |



Table A3: Baseline Parameters (continued)

| Parameter | Description | Units | Value |
|---|---|---|---|
| $\theta^n$ | Fertilizer Conversion Factor | Tton / Ttoe | 1.071 |
| $c^{o,n}$ | Fertilizer Conversion Cost | share of $X_t^O$ | 0.0021 |
| $\theta^{b1}$ | 1G Biofuels Conversion Rate | toe/ton | 0.283 |
| $\theta^{b2}$ | 2G Biofuels Conversion Rate | toe/ton | 0.467 |
| $K$ | 2G Biofuels Fixed Factor Index | | 0.005 |
| $c^{o,b1}$ | 1G Biofuels Conversion Cost | share of $X_t^O$ | 0.00025 |
| $c^{o,b2}$ | 2G Biofuels Conversion Cost | share of $X_t^O$ | 0.00033 |
| $a^n$ | Share of Agricultural Land in CES function | | 0.55 |
| $\rho_n$ | CES Parameter for Agricultural Land and Fertilizers | | 0.123 |
| $A_0$ | Crop Technology Index in 2004 | | 13.89 |
| $\kappa^c$ | Logistic Growth Rate of Crop Technology Index | | 0.025 |
| $c^{o,c}$ | Food Crop Production Cost | share of $X_t^O$ | 0.016 |
| $\theta_0^{c,b2}$ | 2G Biofuels Crop Technology Index in 2004 | | 14.89 |
| $\kappa_{b2}$ | 2G Biofuels Fixed Factor Decay Rate | | 0.05 |
| $\alpha^{b2}$ | Fixed Factor Cost Share in 2G Biofuels Production | | 0.6 |
| $\rho_{b2}$ | CES Parameter for Fixed Factor and Agr. Land | | -1.5 |
| $c^{o,c}$ | 2G Biofuels Crops Production Cost | share of $X_t^O$ | 0.022 |
| $\theta^P$ | Livestock Technology Index in 2004 | | 0.69 |
| $a^l$ | Share of Pasture Land in CES function | | 0.35 |
| $\rho_l$ | CES Parameter for Pasture Land and Feed | | -0.33 |
| $c^{o,l}$ | Livestock Production Cost | share of $X_t^O$ | 0.0055 |
| $\psi_a$ | Merchantable Timber Yield Parameter 1 | | 5.62 |
| $\psi_b$ | Merchantable Timber Yield Parameter 2 | | 76.5 |
| $\overline{v}$ | Minimum Age for Merchantable Timber | Years | 11 |
| $\kappa_v^w$ | Timber Yield Gains of Vintage $v$ | Share of Yield 0 | 0.011 |
| $c^p$ | Forest Planting Cost | share of $X_t^O$ | 0.0001 |
| $c^{o,w}$ | Forest Harvesting Cost | share of $X_t^O$ | 0.0021 |
| *Final Goods and Services* | | | |
| $\theta_0^f$ | Food Processing Technology Index in 2004 | | 1.5 |
| $\kappa^f$ | Food Processing Technology Index Growth Rate | | 0.0225 |
| $c^{o,f}$ | Food Processing Cost | share of $X_t^O$ | 0.015 |
| $\theta_0^l$ | Livestock Processing Technology Index in 2004 | | 1.7 |
| $\kappa^l$ | Livestock Processing Technology Growth Rate | | 0.0025 |
| $c^{o,y_l}$ | Livestock Processing Cost | share of $X_t^O$ | 0.0068 |





Table A3: Baseline Parameters (continued)

| Parameter | Description | Units | Value |
|---|---|---|---|
| $\theta_0^e$ | Energy Technology Index in 2004 | | 1.195 |
| $\kappa^e$ | Energy Technology Index Growth Rate | | 0.0225 |
| $\rho_e$ | CES Parameter for Petroleum and Biofuels | | 0.5 |
| $\alpha^e$ | Share of Biofuels in CES Function | | 0.09 |
| $\theta_0^{y_w}$ | Timber Processing Technology Index in 2004 | | 1.52 |
| $\kappa^{y_w}$ | Timber Processing Technology Growth Rate | | 0.0225 |
| $c^{o,y_w}$ | Timber Processing Cost | share of $X_t^O$ | 0.0224 |
| $\theta^r$ | Ecosystem Services Technology Index | | 0.71 |
| $\alpha^{A,r}$ | Share of Agricultural Land in CES Function | | 0.02 |
| $\alpha^{P,r}$ | Share of Pasture Land in CES Function | | 0.14 |
| $\alpha^{C,r}$ | Share of Managed Forest Lands in CES Function | | 0.26 |
| $\rho_r$ | CES Parameter for Ecosystem Services | | 0.123 |
| $\theta^R$ | Effectiveness Index of Protected Lands | | 10 |
| $c^{o,r}$ | Cost of Recreation Services | | 0.0296 |
| $\theta_0^o$ | Total factor Productivity Index in 2004 | | 1.854 |
| $\kappa^o$ | Total Factor Index Growth Rate | | 0.0225 |

*Preferences and Welfare*

| | | | |
|---|---|---|---|
| $\alpha_f$ | AIDADS Marginal Budget Share at Subsistence Income for Services from Processed Food | | 0.189 |
| $\alpha_l$ | AIDADS Marginal Budget Share at Subsistence Income for Services from Processed Livestock | | 0.035 |
| $\alpha_e$ | AIDADS Marginal Budget Share at Subsistence Income for Energy Services | | 0.112 |
| $\alpha_w$ | AIDADS Marginal Budget Share at Subsistence Income for Services from Processed Timber | | 0.036 |
| $\alpha_r$ | AIDADS Marginal Budget Share at Subsistence Income for Ecosystem Services | | 0.049 |
| $\alpha_o$ | AIDADS Marginal Budget Share at Subsistence Income for Other Goods and Services | | 0.579 |





Table A3: Baseline Parameters (continued)

| Parameter | Description | Units | Value |
|---|---|---|---|
| $\beta_f$ | AIDADS Marginal Budget Share at High Income for Services from Processed Food | | 0.028 |
| $\beta_l$ | AIDADS Marginal Budget Share at High Income for Services from Processed Livestock | | 0.011 |
| $\beta_e$ | AIDADS Marginal Budget Share at High Income for Energy Services | | 0.049 |
| $\beta_w$ | AIDADS Marginal Budget Share at High Income for Services from Processed Timber | | 0.032 |
| $\beta_r$ | AIDADS Marginal Budget Share at High Income for Ecosystem Services | | 0.104 |
| $\beta_o$ | AIDADS Marginal Budget Share at High Income for Other Goods and Services | | 0.776 |
| $\gamma^f$ | AIDADS Subsistence Parameter for Processed Food | | 0.45 |
| $\gamma^l$ | AIDADS Subsistence Parameter for Processed Livestock | | 0.003 |
| $\gamma^e$ | AIDADS Subsistence Parameter for Energy Services | | 0.026 |
| $\gamma^w$ | AIDADS Subsistence Parameter for Processed Timber Products | | 0.027 |
| $\gamma^r$ | AIDADS Subsistence Parameter for Ecosystem Services | | 0.028 |
| $\gamma^o$ | AIDADS Subsistence Parameter For Other Goods and Services | | 0.346 |
| $\gamma$ | Risk Aversion Parameter | | 2 |
| $\delta$ | Social Discount Rate | | 0.95 |



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
