# Peer review of "Assessing Effects of Climate and Technology Uncertainties in Large Natural Resource Allocation Problems"

_EGUsphere, 2022_

## Author Response (AR1)

*Reviewer 1*

*Firstly, the SCEQ algorithm plays a crucial role in the paper, but it is only presented in the appendix and not in the methods section.* **The authors should introduce the algorithm in the methods section** *and, more importantly,* **explain how it has been integrated into the FABLE model**.

We expanded the main body of the manuscript (see lines 235-295 of the revised manuscript) to introduce the algorithm and explain its integration into the FABLE model.

*Similarly, the calibration of the yield development stages is explained only conceptually in the methods section and not specifically for this study.* **The appendix contains relevant details that should be part of the main text.** *Additionally,* **statements in the methods and appendix sections appear contradictory,** *and the authors should clarify the approach they used. While the methods section says that the authors* **"use the results of Rosenzweig et al. (2014)"**, *the appendix says that the authors* **"follow the approach of Rosenzweig et al."** *to process the data, which sounds like slightly different data was used.*

In the revised manuscript (lines 450 and 695), we clarified that climate change impacts on global crop productivity were estimated based on the crop modeling results published by Rosenzweig et al. 2014. That is we use the original data from four crop models forced by five global climate models (see Fig. A1). Since the calibration of yield development stages is based on the previously published data, we chose not to incorporate this material in the manuscript's main text but rather leave it in the appendix.

*The discussion section mostly analyzes model output and feels unrelated to the rest of the paper. The authors* **should either show the model results in the abstract and conclusions and place them in the context of other studies** *or* **put less emphasis on the model output discussion and focus on the paper's main findings**. *The authors should provide* **a more in-depth discussion and justification for the main claim that scenario analysis can overstate the magnitude of expected land conversion under uncertain crop yields**. *Specifically, the authors should explain* **what this overestimation implies and how to interpret it**.

We have completely rewritten the discussion sections 5.1 and 5.2 by (i) shortening the discussion of the model's output as this is primarily a methodological paper, and also to avoid unnecessary overlap with Steinbuks and Hertel (2016), and (ii) providing a more in-depth discussion of stochastic model runs along with more extensive justification for the main claim that scenario analysis can overstate the magnitude of expected land conversion under uncertain crop yields (see e.g., lines 340-250 and 360-370 of the revised manuscript).

*Additionally, the* **authors should discuss the robustness of that finding**. *If I am not mistaken* **the reduced range should be a direct consequence of the user states in the Markov chain** *and its implication that* **the stochastic model "knows" that being already in the best/worst stage the situation cannot get any better/worse in the future**. *It should be critically discussed* **whether this assumption about a bounded solution space is realistic** *or* **might actually lead to an underestimation of the range of potential land conversions under uncertainty**.

We have addressed this concern in footnotes 7 "As rightly noted by an anonymous reviewer, bounded solution space implied by the limited number of states in the Markov chain structure can lead to a reduced solution range. We believe the assumption about a bounded solution space is justified in this context since extreme scenarios in the deterministic analysis should also be unbounded when the best and worst states are unbounded." and 11 "Since the SCEQ algorithm is based on simulation, additional simulations could lead to a wider range, and our current solution could underestimate the range in comparison with the range from all possible simulation paths. The difference will, however, be small and won't affect the economic significance of the main findings."

*The code and data availability section does not provide sufficient information to make the analysis of the paper reproducible. To achieve reproducibility, **the section should provide access to the model source code** used for the analysis **and the input data to the model**. If access to the source code and/or data cannot be made available to the public, this should be explicitly stated and justified in the section. **Code and data should be at least available to the reviewers.***

We provided the model code to the reviewers along with the submission. We also uploaded the code to Github depositary, available at the following link: https://github.com/jsteinbuks/stfable. We have accordingly updated the data and code availability statement in the revised manuscript.

*Specific comments:*

*p3.l83ff: **The wording feels a bit harsh** ("fail to account for") and it also creates in my perspective too high expectations for this paper. Like the other mentioned studies ignored uncertainties in yields this study is ignoring uncertainties the other paper considered. This study does not supersede previous studies but it instead expands the range of uncertainty studies.*

We have revised this wording as "they are less effective in quantifying the effect of uncertainty in the supply and demand drivers in more complex settings, such as, e.g., optimal allocation of multiple competing land resources in the long run" (line 85).

*p7.l184ff: The statement about incomplete coverage of GHG emissions might require some more context. I believe it might be put here to justify that a fixed RCP scenario can be used for the simulations without ensuring that total GHG emissions are in line with this scenario. **Please explain in the text why this consideration is relevant**. In addition, **please also mention and justify in the methods and/or introduction section that RCP6 is used**. Currently this is just mentioned in the appendix with no justification provided. Besides RCP it is also unclear on **what other scenario assumptions (e.g. SSP, mitigation policies) the simulations are based on,** or to **which scenario the simulations can be best compared to. Can the simulations be understood as a business as usual scenario with very limited to none mitigation efforts**?*

We have addressed this concern in the introduction section of the revised manuscript (line 65): "Following Rosenzweig et al. (2014), we use projections from climate and crop simulation models under Representative Concentration Pathways 6.0W/m2 (RCP6) GHG forcing scenario (Moss et al., 2008), as well as the survey of recent agro-economic and biophysical studies to calibrate the index …", and section 3 (lines 190-195 and footnote 6): "Since FABLE is a partial equilibrium model without an embedded climate module, we cannot directly capture all sources of GHG emissions and endogenize their effect on global crop yields. Instead, to ensure the simulation results' comparability with the structural

parameters (e.g., demographic and economic growth, the rate of technological change) of the FABLE model, we select four crop simulation model runs under the RCP6 GHG forcing scenario" […] "FABLE model baseline assumes no climate regulations and other GHG mitigation measures required to achieve RCP4.5 or lower radiative forcing values, whereas realism of RCP8.5 as the 'business-as-usual' scenario has been questioned by the literature (Hausfather and Peters, 2020)."

*p7.l188ff: I was surprised to see the yield states having such a strongly negative bias with states of +15, +2, -15, -19, and -36 percent (are these percentage differences over the simulation period?). Given the details in the appendix this mainly seems to come from 1. having PEGASUS as a rather pessimistic model in the game and 2. considering also runs with deactivated CO2 fertilization effect. As these choices have a critical impact on the final outcomes of the model these choices require some justification in the text. It is also unclear* **why not a more recent analysis such as Jägermeyr 2021 ("Climate impacts on global agriculture emerge earlier in new generation of climate and crop models" Nature Food)** *was taken as point of reference.*

The percentage changes are relative to the reference period 1971 to 2004, which we explain in the revised manuscript, lines 195-200.

This study was designed before updated crop yield projections from Jägermeyr et al. 2021 were available. Future work will make use of these newer simulations. Regarding individual model performance, we believe that each model that passes a benchmarking test qualifies as a stand-alone data point and is thus included in the ensemble mean. It is true that there are large uncertainties across different models, which are largely associated with the $CO_2$ fertilization effect in high-emission climate change scenarios. The $CO_2$ effect is reduced to some degree in the newer GGCMI Phase 3 simulations based on CMIP6, which will benefit follow-on studies. We acknowledge this point in footnote 5: "More recent projections using ensembles of latest-generation crop and climate models find larger uncertainty of climate impacts on major crop yields (Jägermeyr et al., 2021). Unfortunately, these data were not available at the time of the research. Our results should therefore be taken as conservative estimates of the impacts of climate uncertainties on crop yields."

*p11. I am not sure if it helps to show the difference between deterministic and stochastic scenarios on the right hand side.* **Instead I could imagine that showing just the same plot as on the left but for the stochastic runs might ease the comparison between the two. To put the results into context it would be helpful to have the historic development of the variables shown in the plots as well**. *Otherwise it is difficult to evaluate the results.*

Unfortunately, for some figures, the difference between deterministic and stochastic scenarios is too small relative to their absolute magnitude, so it will be difficult to grasp for the reader if illustrated using the same type of plot as on the left. We prefer to keep the illustration as is.

*p14.l338ff:* **Was this strong demand-side elasticity to be expected? Is this in line with other studies?**

Our AIDADS demand system is designed to encompass consumption behavior across a wide range of incomes (Rimmer and Powell, 1996). This is essential for a dynamic model of the global economy. We have estimated three key parameters for each commodity category – the subsistence level of consumption, the marginal budget share at very low (subsistence) income, and the marginal budget share at very high levels of income. The former two are large for food products. However, as households

become wealthy, the marginal budget share for food items becomes very small, approaching zero for very high incomes. In this application, as households become wealthier, the subsistence share becomes very small, and households' demand response becomes larger. We acknowledge this point in footnote 12 of the revised manuscript.

*p18.l387: I did not find a description in the text of how the high-resolution outputs were plugged into the FABLE model.* **Is the FABLE model spatial explicit?**

The FABLE model is not spatially explicit, so we had to aggregate gridded crop yields, weighting by the size of crop output per grid cell. We clarify this point in lines 695-700 of the revised manuscript.

*Reviewer 2 (Xin Zhao)*

*Deterministic vs. perfect foresight*

**The related concepts need to be clarified**. *It seems the study assumes the deterministic model to be perfect foresight, which may not always be the case. By perfect foresight, is it describing agricultural producers or land use decision-makers? However, in the stochastic model, is there imperfect foresight for producers, e.g., rational expectations?*

Perfect foresight describes a global planner optimally allocating the model's land uses. The stochastic model still has the same global planner but now with imperfect foresight, indeed rational expectations. We clarify this point in the revised manuscript, lines 70-75: "We simulate the results of the model where the global planner optimally allocates land uses under the perfect foresight of different realizations of the crop productivity index, focusing our attention on the current century. We then compare and contrast them with the results of the dynamic stochastic model, where the global planner has rational expectations about uncertain crop yields brought to the model's optimization stage."

*A relative question is* **how the expected utility is calculated**, *e.g., line 575. Are expectation schemes assumed for the calculation?*

The expected utility is just a sum of utilities in each state time the probability of each state in a given period, where exogenous states evolve stochastically over time according to a Markov process with time-varying transition probabilities defined in Appendix C. We clarify this in the revised manuscript, lines 615-620.

*How uncertainty affects the optimal path of land use?*

*The main contribution/goal of this study was to showcase properly accounting for the uncertainty that could affect decision-maker behavior. However,* **the logic behind this was not thoroughly communicated**.

*Some documentation of the parameters used in land conversion cost functions (Eq. D33 – 36) could be useful, as they seem to be relevant to land supply/transformation elasticities.*

To our knowledge, there are no empirical studies estimating the magnitudes of long-term adjustment costs in land conversion problems. We, therefore, choose to calibrate these parameters to match historical land conversion patterns. We clarify this in the revised manuscript, lines 505-510.

*Is rental profit a factor in land allocation? And is the landowner risk-averse?*

We assume no rental profits as those are fully redistributed by global planners back to consumers of land-use goods and services. We clarify this in the revised manuscript, lines 470-475. The global land planner is indeed risk averse as we explain in Appendix A4.

*What about **endogenous market fluctuation by wrong market price expectations**, e.g., cobweb models?*

In this model, we assume that agents know the underlying distribution of crop productivities, so the market expectations are, on average, accurate. We do not encounter cobweb-type behavior in this model.

*FABLE and climate/crop models*

*It seems the crop production function in FABLE did not include "other primary inputs" which are available in the model. However, only including fertilizer and land in production, **do you assume the rest of the costs are absorbed by land profit (assuming there is zero profit condition)**?*

We treat these costs as exogenous and assume they have an 'iceberg' representation, i.e., they are subtracted from the gross output of land-based goods and services. We clarify this in the revised manuscript, lines 550-555.

*How was ecosystem service valued in the model? E.g., it is included in the utility function and supply by land. But how was it valued in data and parameterized in modeling.*

Calibration details are available in section B.1.12 of supplementary materials to Steinbuks and Hertel (2016), accessible at https://static-content.springer.com/esm/art%3A10.1007%2Fs10640-014-9848-y/MediaObjects/10640_2014_9848_MOESM1_ESM.pdf

*How many crops are included in FABLE? Was there a mapping between crop models and FABLE? E.g., are there climate impacts on bioenergy crops?*

The FABLE model has one global crop, which is an output-weighted composite of four major crops: wheat, rice, corn, and soybeans. We assume that food crops are converted to first-generation biofuels so climate impacts on first-generation biofuels crops are the same as on food crops. The FABLE model assumes that second-generation biofuel crops' yields are not affected by climate change (see lines 415-410 of the revised manuscript).

*Climate scenarios are not clear. Rosenzweig et al. (2014) used RCP 8.5 scenarios, which were from ISIMIP fast track data. However, it is stated RCP 6.0 is used in this study. **Were those data from ISIMIP2b database?** Please include this information in the main text*

Rosenzweig et al. 2014 (ISIMIP fast track) used simulations for RCP2.6, RCP4.0, RCP6.0, and RCP8.5. Here use the results for RCP6.0. Results from ISIMIP2b are not used. We clarify this in the introduction section of the revised manuscript (line 65): "Following Rosenzweig et al. (2014), we use projections from climate and crop simulation models under Representative Concentration Pathways 6.0W/m2 (RCP6) GHG forcing scenario (Moss et al., 2008), as well as the survey of recent agro-economic and biophysical studies to calibrate the index …", and section 3 (lines 190-195 and footnote 6): "Since FABLE is a partial equilibrium model without an embedded climate module, we cannot directly capture all sources of GHG emissions and endogenize their effect on global crop yields. Instead, to ensure the simulation results' comparability with the structural parameters (e.g., demographic and economic growth, the rate of technological change) of the FABLE model, we select four crop simulation model runs under the RCP6 GHG forcing scenario" […] "FABLE model baseline assumes no climate regulations and other GHG mitigation measures required to achieve RCP4.5 or lower radiative forcing values, whereas realism of RCP8.5 as the 'business-as-usual' scenario has been questioned by the literature (Hausfather and Peters, 2020)."

*Line 705, **does FABLE have a climate model and provide a reference projection of RCP 6.0?***

The FABLE model doesn't have an internal climate module as this is not an integrated assessment model. Instead, we rely on projections from five global climate models based on RCP 6.0 scenario. We clarify this in the revised manuscript, lines 190-195.

*Results*

*It might be useful to **describe the reference scenario of FABLE**, e.g., the one with no climate impacts.*

We will briefly describe this scenario in the revised manuscript (see our response to reviewer 1) but following their recommendation will avoid the in-depth description previously published in Steinbuks and Hertel (2016).

*Overall, **the communication of the results can be improved**. For example, it seems the comparison of optimistic-pessimistic range between deterministic and stochastic is important. **A figure focusing on the comparison, e.g., in the same unit (Mha), could be useful.***

We have revised the right-hand sides of Figures 3 and 4 to reflect these suggestions.

***In addition to land, market prices could also be important**, e.g., will there be higher price variation?*

Since this is a social planner's problem all prices are effective shadow prices, which are determined endogenously by the model.

*Importance:*

*Minor comments/questions:*

*Abstract: "The scenario analysis can thus significantly overstate the magnitude of expected land conversion under uncertain crop yields." **Not very clear by "magnitude" and why "expected".** Maybe*

*just a sentence highlighting the importance of incorporating uncertainty into the determination of the optimal path of natural resource use?*

We have revised this sentence as "This highlights the importance of incorporating uncertainty in the model's optimization stage to determine optimal paths of natural resource uses."

*Line 119, "they are typically left out of most contemporary analyses of global land use change," this might be true 10 years ago. But there has been growing interest in including all land in the modeling.* **E.g., does unmanaged forest has value in the base year?**

This point is well-taken. We have modified the text to read as follows (footnote 3):

"… they have historically been neglected in economic models of global land use change. More recently, these natural lands have been incorporated via location-specific supply curves depicting the potential for bringing these lands into commercial production (REF MAGNET model: https://www.magnet-model.eu/model/ ). However, the ecosystem services provided by these lands are not explicitly valued as they are in the FABLE model, where they are explicitly included in the utility function."

**Lines around 190, are those percent changes of yield global median values?**

No, these are changes relative to the historical trend over 1971-2004 (calibrated model baseline), see lines 195-200 of the revised manuscript.

**Line 210, J1 can only move up, and J5 can only move down?** *And is such move per model period (5 years) or per annum?*

Yes, J1 can only move up, and J5 can only move down, and such a move is per model period (5 years). We clarify this in the revised manuscript, lines 215-220.

**Line 575, the notations of Section A4?**

"The notations of section 4", see the revised manuscript, around line 625.

*The references are somewhat dated. Consider updating if appropriate.*

**FYI, we have a relevant recent study: "Global agricultural responses to interannual climate and biophysical variability."** *We used adaptative expectations for both price and yield for Ag producers to make land allocation and production decisions. We had similar results that land use change variation became much smaller compared to perfect foresight because of the slower adjustments under imperfect foresight. But market price variations increased.*

We have revised the manuscript as follows to incorporate novel literature (see lines 95-100: "Zhao et al. (2021) compare models with adaptative expectations and perfect foresight assumptions for both price and yield for agricultural producers to make land allocation and production decisions. Zhao et al. (2021) find similar results that land use change variation becomes much smaller than in the perfect foresight model, which allows for faster land use adjustments while market price variations increase. Unlike our paper, Zhao et al. (2021) do not explicitly incorporate uncertainty in the model's optimization stage."

---

## Author Response (AR2)

Reviewer 1

*I thank the authors for their effort in revising the paper. All specific comments have been adequately addressed. Also the SCEQ algorithm is now explained in the manuscript. Concerning code availability I welcome that the authors made their code available via GitHub. However, I would like to point out that the chosen format to upload it as a zip archive to GitHub is unusual. As it fulfills my initial request to make the code available it could stay as it is now, but I would nevertheless recommend to convert it to a more conventional format of having the model uploaded in an uncompressed form and have the model archived in a public archiving service (e.g. Zenodo) to ensure that the code is also available for download in the future.*

**We have uploaded the model solution code to Zenodo (see https://zenodo.org/records/10014997).**

*The remaining issue where I still feel that the authors did not address my concerns appropriately is the discussion of the robustness of the main finding of the paper (overestimation of land conversion ranges in deterministic models). Instead of adding a discussion of the robustness of finding the authors added a footnote claiming that a unbounded solution space only would make sense if the deterministic scenarios would be unbounded as well.*

**We have removed this footnote in the revised manuscript.**

*Firstly, I am not sure how an unbounded deterministic scenario should look like given that due to its deterministic nature all conditions are prescribed. Hence, the authors seem to claim that every comparison with a deterministic scenario requires a bounded solution space for the corresponding stochastic scenario. It is not clear to me where this conclusion comes from or how it could be justified.*

*Secondly, I think that the authors have not fully grasped the point I was trying to make in my last response so I will try to rephrase and repeat it: For simplicity, let's just have a look at the comparison in Figure 3a) between stochastic pessimistic and deterministic pessimistic where we can clearly see the difference between the two with the stochastic pessimistic scenario showing less agricultural land expansion than the deterministic scenario. In combination with the optimistic scenarios that leads to the claim in the paper that the deterministic analysis leads to an overestimation in the range. So the question is, why does the stochastic approach leads to a smaller range and the answer in the paper is that this happens because it is stochastic. I clearly disagree here. The reason is not that it is stochastic, but that the stochastic scenario works with a bound solution space.*

*Comparing both pessimistic scenarios we have on one hand a deterministic scenario which computes a world in which everything is bad and we know for certain that it stays that bad. At the same time the stochastic scenario computes a world in which we start in a pretty bad state but we know that it will either stay that bad or get better but under no circumstances get worse! That means that this scenario is by definition (due to the bound solution space) a better world than the one computed in the deterministic pessimistic scenarios. Hence, we also end up with less agricultural area in use. In case of an unbounded solution space the model could not make this assumption and would need to prepare for shifts in either direction (for better or worse). How such a scenario behaves we can see quite well looking at the average. Here the probability to get better or worse is identical and the findings of the stochastic and deterministic scenarios more or less agree.*

**Thanks for this clarification. We have added the following sentences on p. 17, line 375 of the revised manuscript to clarify this point further: "Compared with the deterministic model under the pessimistic (or optimistic) scenario, the social optimum in the stochastic model requires a smaller (or greater) conversion of other types of land to cropland. This is because when the current state of the crop technology index is the worst (best), its future states cannot be worse (better) and have a nonzero probability of being better (worse). The expected future yields will then be better (worse) than the deterministic-pessimistic (optimistic) scenario. As the size of expected crop yields affects the magnitude of the land conversion decisions, the range of stochastic model solutions for agricultural land will be smaller than the range between the most extreme deterministic model solutions.".**

*Hence, one can argue that a bound solution space will lead to a smaller range in a stochastic approach compared to the range we get if we run the bounds in a deterministic model, but this is only true for a bound solution space. The question is whether a bound solution space is a reasonable assumption here. I would not think so. Given that this is a more conceptual analysis it is okay to use a potentially unrealistic assumption but it is absolutely critical to properly report and discuss it. In particular the required conditions under which this result holds true need to be clearly mentioned as the results are otherwise misleading.*

**Thanks for bringing this issue to our attention again. We have added the following sentences on p. 17, line 375 of the revised manuscript to address this concern: "Note this result may not hold if the model solution space is unbounded. This concern doesn't apply to the stochastic FABLE model because (i) the model's time horizon is finite; (ii) the crop technology shocks are discrete and finite (hence bounded) based on scientific projections used for the model's calibration; (iii) all of the model's state variables (land and fossil fuel resources) are bounded because the total land and the total fossil fuel resources are finite; and (iv) we impose bounds on model decision variables based on the theory of economic dynamics (Barro and Sala-i-Martin, 2004), such as strictly positive and finite consumption and output of land-based goods and services; land conversion cannot exceed the total supply of land). The solution space, therefore, must also be bounded because the extent of movement of optimal land uses in any direction is limited by the constraints mentioned above."**

Reviewer 2

*1. About climate impact scenarios from ISIMIP (fast-track) or Roseweig et al. (2014), was there a reason for not using more recent scenarios? I believe Jonas Jaegermeyr, a coauthor of the paper, has led studies/experiments of the more recent ISIMIP rounds (e.g., ISIMIP 3b). I think it would be great to provide some clarifications. Using fast-track data seems fine for testing the model, but I think audiences would like to know to what extent more recent data could affect the results.*

**Unfortunately, these data were unavailable when the manuscript was submitted to GSM (it took a long time for the journal editors to find reviewers). Given the methodological scope of the paper and very lengthy revisions associated with revising model baseline and climate yield shocks, we chose to leave this exercise for future, more policy-oriented research.**

*2. In your response, you mentioned: "Since this is a social planner's problem all prices are effective shadow prices, which are determined endogenously by the model." I am confused by the shadow prices here. The model has both demand and supply. Isn't that prices are variable solved? I would appreciate more clarifications as to why market price results cannot be provided.*

The model solves for optimal quantities (i.e., different allocation of land uses in GHa) under the supply (e.g., total land and fuel resources) and demand (final consumption of land-based goods and services entering utility function) constraints, both in quantities. The prices don't explicitly enter the model and are calculated as Lagrange multipliers (or shadow prices) for respective resource constraints.

---

## Author Response (AR3)

*Editor*

*The revised manuscript has been seen by referee #1 once more. I concur with their assessment that the revisions insufficiently address the outstanding concerns. The assumption that crop technology parameters are (a) bounded to the most pessimistic / optimistic case of the deterministic model (which are not necessarily the maximum / minimum values from a biophysical perspective, and (b) not auto-correlated along the time axis (low crop yields at time t make it less likely that yields are high at time t+1) introduces artifacts that affect the core conclusions drawn in the papers abstract and conclusions.*

The assumption (a) that crop technology parameters are bounded is necessary to avoid completely unrealistic cases when crop productivity becomes infinitely large or small and is consistent with biophysical modeling studies cited in our paper. As regards the assumption (b) the crop technology parameters are, in fact auto-correlated along the time axis in the stochastic FABLE model.

*To remedy this problem, the authors should revise the underlying model. If there are fundamental reasons preventing you from adjusting your model, the interpretation of the results as well as abstracts and conclusions must be thoroughly revised.*

In light of the above, we don't think the model itself needs any revisions. Instead, we have made further clarifications about the validity of assumptions (a) and (b) in the revised manuscript. As regards the assumption (a), we have made it clear in the abstract, introduction, the model description, and the conclusions part, that our model has a bounded solution space, which is a standard assumption in economic modeling of natural resources, and that our shocks are bounded to avoid unrealistic realizations of crop technology parameters that would prevent finding model solution. Please refer to our response to reviewer 1 for the list of specific changes made. As regards the assumption (b), please note the following sentence on line 293 of the revised manuscript "Observe that the serial correlation of random variables has been captured in their associated transition laws", and the appendix section B3 describing the discretization of autocorrelated shocks.

*Reviewer 1*

*I thank again the authors for adjusting the manuscript based on the made recommendations. The text now contains two paragraphs explaining the linkage between the bound solution space and the reduce range of outcomes. However, the new manuscript still completely ignores that the key findings of the paper are basically invalidated by it. For example the abstract still says that "For the same model parameters, the range of land conversion is considerably smaller for the dynamic stochastic model as compared to deterministic scenario analysis. This highlights the importance of incorporating uncertainty in the model's optimization stage to determine optimal paths of natural resource uses." This is plainly misleading as a) the smaller range is not a consequence of running it with a stochastic model, but a consequence of the bounded solution space (in an unbounded stochastic case the range would be identical) and b) incorrectly deducts that this would highlight the necessity to do stochastic analysis (there are other good reasons for stochastic analysis, but this is not one of them).*

We have revised the abstract, deleting the last sentence, and changing previous sentence to "For the same model parameters and bounded shocks, the range of land conversion is considerably smaller for the dynamic stochastic model than for deterministic scenario analysis." Indeed our findings will not necessary hold when shocks are unbounded.

*In addition, to justify that the approach of a bound solution space by the explicit assumption of a bound solution space in the FABLE model is somehow questionable. It just means that the yield assumptions in the model are consistent to the other assumptions of the model, it does not mean that it is a sensible approach overall. The main question is whether a bound solution space is a reasonable approach to model reality or if it is not and there are good reasons to question this assumption.*

We have addressed this concern by adding the following sentences on line 107 of the revised manuscript: "Similar to other models in this class, the FABLE model has a bounded solution space, as all these models are theoretically shown to have equilibrium paths (Stokey, 1989). This assumption is important because it is often impossible to prove that in the presence of unbounded solution space, the stochastic model has a finite solution. For example, Weitzman's dismal theorem (Weitzman, 2010) shows that a fat-tail damage function with an infinite upper bound leads to an infinite risk premium, but a numerical truncation to finite support will always have a finite risk premium. So, assuming the bounded solution space is necessary for avoiding the potential qualitative inconsistencies between their theoretical and numerical results. We further discuss the assumption of the bounded solution space in the FABLE model in section 5.

*To bring this manuscript into a form acceptable for publication it would be absolutely important not only to mention the shortcomings of this methodology, but also to get rid of all these incorrect conclusions in abstract, discussion and results. At the moment it mentions the shortcomings at some point but downplays them and completely ignores them in the rest of the manuscript.*

In addition to changing abstract we have added the following sentences in the introduction and conclusions sections:

Line 60: "To ensure consistency between theoretical and numerical model solutions, we assume the bounded solution space. As we show below, this assumption is well justified for economic models of large natural resource allocation problems, including the FABLE model."

Line 72: "We then compare and contrast them with the results of the dynamic stochastic model, where the global planner has rational expectations about crop yields subject to bounded autocorrelated climate shocks"

Line 78: "This result indicates that when the climate shocks are bounded, the scenario analysis may significantly overstate the expected agricultural land conversion magnitude under uncertain crop yields."

Line 464: "Similar to other dynamic economic models, it assumes a bounded solution space,

excluding unrealistic scenarios of infinitely low or high crop productivity."

*Given that these concerns were already raised in the last two rounds of review I am a bit surprised and concerned that this still has not been properly addressed by now.*

We hope this last iteration satisfactory addresses your remaining concerns.